# Morphology and ultrastructure of external sense organs of *Drosophila* larvae

**Vincent Richter[1], Anna Rist[2], Georg Kislinger[2], Michael Laumann[3], Andreas Schoofs[4], Anton Miroschnikow[4], Michael J Pankratz[4], Albert Cardona[5], Andreas S Thum[1,2,6]\***

[1]Department of Genetics, Leipzig University, Institute for Biology, Leipzig, Germany; [2]Department of Biology, University of Konstanz, Konstanz, Germany; [3]Electron Microscopy Center, University of Konstanz, Konstanz, Germany; [4]Department of Molecular Brain Physiology and Behavior, LIMES Institute, University of Bonn, Bonn, Germany; [5]Department of Physiology, Development and Neuroscience, University of Cambridge, Cambridge, United Kingdom; [6]German Centre for Integrative Biodiversity Research (iDiv) Halle-Jena-Leipzig, Leipzig, Germany

## eLife assessment

The manuscript from Richter et al. is a very thorough anatomical description of the external sensory organs in *Drosophila* larvae. It represents a **fundamental** step forward for sensory physiology, and provides a tool for investigating the relationship between the structure and function of sensory organs. Using improved electron microscopy analysis and digital modelling, the authors provide **compelling** evidence that form the basis for further molecular and functional studies to decipher the sensory strategies used by larvae to navigate through their environment.

**\*For correspondence:**
andreas.thum@uni-leipzig.de

**Abstract** Sensory perception is the ability through which an organism is able to process sensory stimuli from the environment. This stimulus is transmitted from the peripheral sensory organs to the central nervous system, where it is interpreted. *Drosophila melanogaster* larvae possess peripheral sense organs on their head, thoracic, and abdominal segments. These are specialized to receive diverse environmental information, such as olfactory, gustatory, temperature, or mechanosensory signals. In this work, we complete the description of the morphology of external larval sensilla and provide a comprehensive map of the ultrastructure of the different types of sensilla that comprise them. This was achieved by 3D electron microscopic analysis of partial and whole body volumes, which contain high-resolution and complete three-dimensional data of the anatomy of the sensilla and adjacent ganglia. Our analysis revealed three main types of sensilla on thoracic and abdominal segments: the papilla sensillum, the hair sensillum, and the knob sensillum. They occur solitary or organized in compound sensilla such as the thoracic keilin's organ or the terminal sensory cones. We present a spatial map defining these sensilla by their position on thoracic and abdominal segments. Furthermore, we identify and name the sensilla at the larval head and the last fused abdominal segments. We show that mechanosensation dominates in the larval peripheral nervous system, as most sensilla have corresponding structural properties. The result of this work, the construction of a complete structural and neuronal map of the external larval sensilla, provides the basis for following molecular and functional studies to understand which sensory strategies the *Drosophila* larva employs to orient itself in its natural environment.

## Introduction

*Drosophila melanogaster* larvae have become a favored model organism for studying the principles of sensory perception (reviewed in *Gerber and Stocker, 2007*; *Melcher et al., 2007*; *Apostolopoulou et al., 2015*; *Joseph and Carlson, 2015*; *Rimal and Lee, 2018*; *Widmann et al., 2018*; *Thum and Gerber, 2019*). It has been shown that larvae actively use sensory cues of different modalities to navigate through their environment. Thermo-, photo-, chemo-, and mechanosensory information is perceived and processed by the larvae's peripheral nervous system to optimize their behavior output for favorable conditions (e.g. to find food sources) and to avoid unpleasant or even harmful situations (e.g. predators and deterrent or toxic substances) (*Tracey et al., 2003*; *Gomez-Marin and Louis, 2012*; *Apostolopoulou et al., 2014*; *Klein et al., 2015*; *Ohyama et al., 2015*; *Scholz et al., 2015*; *Apostolopoulou et al., 2016*; *Choi et al., 2016*; *Croset et al., 2016*; *Ni et al., 2016*; *van Giesen et al., 2016*; *Humberg et al., 2018*; *Tastekin et al., 2015*).

An increasing number of studies investigating the neuronal and molecular basis of larval sensory perception has even enabled the identification of the respective sensory cells and receptors for distinct stimuli. Among others, this includes a distinct set of olfactory receptors (Or) (*Fishilevich et al., 2005*; *Kreher et al., 2005*), gustatory receptors (Gr), ionotropic receptors (Ir), pickpocket receptors (ppk), transient receptor potential (TRP) channels, and rhodopsins (Rh). Some receptor genes have been associated with specific sensory functions, for instance, Gr43a (fructose) (*Mishra et al., 2013*), Gr28a (ribonucleosides and RNA) (*Mishra et al., 2018*), Gr33a, Gr66a, and Gr97a (quinine) (*Apostolopoulou et al., 2014*), Gr33a, Gr66a, and Gr93a (caffeine) (*Apostolopoulou et al., 2016*), Ir76b (amino acids) (*Croset et al., 2016*), Ir25a (denatonium) (*van Giesen et al., 2016*), ppk11, and ppk19 (low salt) (*Liu et al., 2003*; *Alves et al., 2014*), ppk23 and ppk29 (pheromones) (*Mast et al., 2014*), painless and ppk (noxious heat) (*Tracey et al., 2003*), ppk and piezo (noxious mechanical stimuli) (*Zhong et al., 2010*; *Kim et al., 2012*), dCIRL and nompC (mechanosensation) (*Scholz et al., 2015*; *Scholz et al., 2017*; *Yan et al., 2013*), Ir68a, Ir93a, Ir25a, and Ir21a (temperature) (*Klein et al., 2015*; *Ni et al., 2016*; *Hernandez-Nunez et al., 2021*), and Rh5 and Rh6 (vision) (*Humberg et al., 2018*). This sensory description is complemented by a large-scale electron microscopic reconstruction of the larval brain (*Winding et al., 2022*). The reconstruction characterizes at the cellular and synaptic level the sensory inputs and initial information processing steps for the olfactory antennal lobe (*Berck et al., 2016*), the optic neuropil (*Larderet et al., 2017*), the gustatory subesophageal zone (*Miroschnikov et al., 2018*) and the nociceptive and mechanosensory brain centers (*Ohyama et al., 2015*; *Jovanic et al., 2016*; *Takagi et al., 2017*; *Burgos et al., 2018*; *Jovanic et al., 2019*; *Masson et al., 2020*).

Despite these advances, the structural basis of sensory perception in *Drosophila* larvae is far from fully elucidated because electron microscopic reconstruction is still limited to the central nervous system. A comparable analysis for the entire peripheral nervous system is not available.

Still, most knowledge on the ultrastructure of larval sensory organs and their sensilla derives from studies of different dipteran species carried out mostly during the 1970s and 1980s (*Hertweck, 1931*; *Thurm, 1964*; *Chu and Axtell, 1971*; *Chu-Wang and Axtell, 1972a*; *Chu-Wang and Axtell, 1972b*; *Zacharuk, 1972*; *Denell and Frederick, 1983*; *Campos-Ortega and Hartenstein, 1985*; *Dambly-Chaudière and Ghysen, 1986*; *Whittle et al., 1986*; *Honda and Ishikawa, 1987*; *Lanfranchi and Belcari, 1990*; *Keil, 1997*). Most of this work was done on different developmental stages of dipteran larvae using different methodical approaches. Current attempts to understand the relationship between the structure and function of larval sensory organs of *Drosophila* are, therefore, hampered by an anatomical description that, on the one hand, is detailed for specific aspects and species but, on the other hand, has gaps for the *Drosophila* larva that are only indirectly addressed from anatomical results of different fly species.

To overcome this limitation and gain precise knowledge of peripheral sensory organ ultrastructure, we have recently analyzed the anatomy of the terminal organ (TO) of the *Drosophila* larva, its major external taste organ (*Rist and Thum, 2017*). This was possible by taking advantage of technical improvements in volume electron microscopy. In particular, we used focused ion beam scanning electron microscopy (FIB-SEM) to gain precise, three-dimensional reconstructions of each of the 14 external sensilla of the TO. FIB-SEM can automatically generate serial images of ultrastructure with superior z-resolution compared to other common volume EM techniques. Extremely thin layers of the specimen are ablated by an ion beam and an image is taken by the SEM after each removed layer (*Helmstaedter et al., 2011*; *Peddie and Collinson, 2014*). A more 'classic' approach is serial

section scanning transmission electron microscopy (ssTEM). In this approach, ultrathin sections of the complete sample are created and then scanned using transmission electron microscopy. These methods have increasingly been used to obtain 3D representations of cellular and even subcellular structures at high resolution (*Peddie and Collinson, 2014*; *Titze and Genoud, 2016*).

Using partial larval 3D volumes based on FIB-SEM and a full larval body volume established via the ssTEM technique (serial sections imaged with a TEM in scanning mode) (*Peale et al., 2024*; *Schoofs et al., 2024*), we have now analyzed the anatomy of the remaining three major head sensory organs, the dorsal organ (DO), the ventral organ (VO), and the labial organ (LO) at ultra-resolution. The three peripheral sensory organs are known to be formed by several sensilla (*Hertweck, 1931*; *Chu and Axtell, 1971*; *Chu-Wang and Axtell, 1972b*; *Kankel, 1980*; *Singh and Singh, 1984*; *Python and Stocker, 2002*). The DO was proven to be the primary larval olfactory organ based on anatomical, molecular, and functional experiments (*Chu and Axtell, 1971*; *Singh and Singh, 1984*; *Heimbeck et al., 1999*; *Oppliger et al., 2000*; *Python and Stocker, 2002*; *Fishilevich et al., 2005*; *Kreher et al., 2005*). Its prominent 'dome' houses 21 olfactory receptor neurons organized in seven triplets that respond to different sets of odors. Less is known about the six peripheral sensory sensilla and their additional roles in thermosensation (*Klein et al., 2015*; *Ni et al., 2016*; *Hernandez-Nunez et al., 2021*) and putatively mechano- and taste sensation (*Chu and Axtell, 1971*; *Singh and Singh, 1984*; *Python and Stocker, 2002*). The VO and the LO are comparatively small sensory organs and have been little noticed in larval anatomical or functional studies (*Chu-Wang and Axtell, 1972b*; *Python and Stocker, 2002*; *Miroschnikow et al., 2018*). They are assumed to serve a mechanosensory and/ or gustatory function.

Another focus of our work was to describe the morphology and ultrastructure of the external sensory organs of the thoracic and abdominal segments. A comparatively simple organization was reported for these. The majority of them consisted of only one sensillum, which was described to be either of the campaniform, basiconic, or trichoid type (*Kankel, 1980*; *Singh and Singh, 1984*; *Campos-Ortega and Hartenstein, 1985*; *Hartenstein, 1988*; *Campos-Ortega and Hartenstein, 1997*). Other studies, however, called campaniform sensilla papilla or pit sensilla. Trichoid sensilla were also called hair sensilla. For the basiconic sensilla, the nomenclature is most diverse, as these are called koelbchen, knob, knob-in-pit, hair-type B, sensory papillae, dorsal or ventral pit, or black sensory organ (*Hertweck, 1931*; *Dambly-Chaudière and Ghysen, 1986*; *Singh and Singh, 1984*; *Rist and Thum, 2017*; *Singh, 1997*; *Kankel, 1980*; *Lohs-Schardin et al., 1979*; *Hartenstein, 1988*; *Sato and Denell, 1985*; *Lewis, 1978*; *Campos-Ortega and Hartenstein, 1985*). In addition, not all sensilla are described and named in the fused first head and last abdominal segments (*Schmidt-Ott et al., 1994*; *Courtney et al., 2000*; *Wipfler et al., 2013*). Accurate classification and nomenclature of the different types of sensilla throughout the larval body – as applied in this work - will, therefore, be useful for future anatomical and functional studies.

## Results

Proper classification of sensilla requires investigation of their external and internal morphology. However, the ultrastructure of scattered and small insect sensilla, like that of *Drosophila* larvae, is challenging to investigate. Fortunately, recent advances in EM technique made it possible to image large regions of tissue, like the entire central nervous system of larval (*Ohyama et al., 2015*; *Schlegel et al., 2016*; *Carreira-Rosario et al., 2018*; *Miroschnikow et al., 2018*; *Winding et al., 2022*) and adult *Drosophila melanogaster* (*Zheng et al., 2018*; *Scheffer et al., 2020*; *Schlegel et al., 2023*), *Caenorhabditis elegans* (*White et al., 1986*; *Cook et al., 2019*), *Ciona intestinalis* (*Ryan et al., 2016*), and the larva of *Platynereis dumerilii* (*Verasztó et al., 2020*), as well as parts of the brain of P*ristionchus pacificus* (*Bumbarger et al., 2013*; *Hong et al., 2019*) and rodents (*Denk and Horstmann, 2004*; *Helmstaedter et al., 2013*; *Motta et al., 2019*). In this data, we reconstructed the external sensory sensilla in a full body first instar EM volume (*Peale et al., 2024*; *Schoofs et al., 2024*) and in volumes of single sensilla of third instar larvae obtained by FIB-SEM. For single sensilla imaging, accuracy and precision are required which are often difficult to realize technically. In order to achieve these technical demands, we took advantage of FIB-SEM in combination with an optimized preparation protocol. This allowed us to exactly target even the smallest sensilla on the larval body wall for subsequent serial slicing and imaging by FIB-SEM.

## Structural organization of *Drosophila* larvae

The body of *Drosophila* larvae is divided into segments (*Figure 1A*): a pseudocephalon (Pce, from now on called 'head,' *Figure 1A'*), three thoracic (T1-T3) and nine abdominal (A1-A9) segments (*Campos-Ortega and Hartenstein, 1997*). The head is the strongly reduced head capsule of the larva, dorsally fused with and partially retracted in the prothorax (*Courtney et al., 2000*; *Wipfler et al., 2013*). The last abdominal segment, A8, is more appropriately named 'anal division' as it is formed by the fusion of at least two abdominal segments A8 and A9 (*Figure 1A''*).

The largest sense organs of *Drosophila* larvae are arranged in pairs on the right and left side of the head (*Figure 1A*, inset left bottom). The DO and the TO are prominently located on the tip of the head lobes (*Chu and Axtell, 1971*; *Chu-Wang and Axtell, 1972a*; *Singh and Singh, 1984*; *Heimbeck et al., 1999*; *Oppliger et al., 2000*; *Python and Stocker, 2002*; *Fishilevich et al., 2005*; *Kreher et al., 2005*; *Rist and Thum, 2017*). The VO and the LO are smaller and their ultrastructure has rarely been studied (*Hertweck, 1931*; *Kankel, 1980*). The VO is situated ventral to the DO/TO complex on the front of the larval head hidden behind rows of cuticle cirri (*Figure 1A'*, Figure 8B). The LO is inconspicuously located below the larval mouth opening (*Figure 1A'*, Figure 9B). In this work, we describe the morphology of the DO, VO, and LO each in a specific result section, and present a conclusive nomenclature of their sensilla (*Figures 2–9*, *Table 1*). The detailed morphology of the TO was described in our previous work (*Rist and Thum, 2017*). The anatomical data is, therefore, not shown again. However, its morphology is part of *Table 1* to complement the description of the external head sensory organs.

Sensory organs located on the thoracic and abdominal segments are of simpler organization mostly consisting of only one sensillum (*Figure 1B*). Different names have been given to these sensilla (*Table 2*). In the present work, we identified three basic types of thoracic and abdominal sensilla and refer to them as papilla (p), hair (h), and knob (k) sensilla (*Figures 1 and 2*, *Figures 10–13*, *Table 2*). This nomenclature corresponds to the one of *Dambly-Chaudière and Ghysen, 1986*, which is most consistent with our findings. Please note that we have translated their term 'kölbchen sensilla' into English as 'knob sensilla.' The term knob or knob-in-pit sensillum was already used in previous studies (*Singh and Singh, 1984*; *Rist and Thum, 2017*; *Singh, 1997*). It allows us to use the same abbreviation (e.g. vk for ventral knob sensillum), which also allows for comparison with these studies. We are aware that the term papilla sensilla is questionable, as the outer and inner morphology resembles the one of the spot sensilla we found in the TO in our previous work (*Rist and Thum, 2017*). The term papilla sensilla is based on findings in light microscope data and does not describe the outer morphology appropriately. Nevertheless, renaming these sensilla would be inconvenient for the comprehension and comparability with already published data and would make it necessary to use different abbreviations.

Papilla, hair, and knob sensilla are in some studies called campaniform, trichoid, and basiconic sensilla, respectively (*Hartenstein, 1988*; *Campos-Ortega and Hartenstein, 1997*; *Green and Hartenstein, 1997*). Of course, these terms also have their validity and are, therefore, listed in *Table 1*. However, since in our view, these terms were structurally more difficult to assign and could lead to confusion from a functional point of view, we prefer the former. Trichoid sensilla in the adult olfactory system of *Drosophila* express odorant receptor genes and thus exert a chemosensory function (*Couto et al., 2005*; *Miller and Carlson, 2010*; *van der Goes van Naters and Carlson, 2007*). In contrast, any potential larval trichoid sensilla most-likely has only a mechanosensory function due to its ultrastructure.

In general, most external sensilla are arranged along the vertical (dorso-ventral) axis close to the middle of each segment (*Figure 1B*). In accordance with previous studies (*Lohs-Schardin et al., 1979*; *Dambly-Chaudière and Ghysen, 1986*; *Green and Hartenstein, 1997*), we found a stereotypic and fixed pattern of these sensilla. This structural consistency allowed us to generate a spatial map defining each sensillum and sense organ on the larval body by their precise position (*Figure 1B*). Furthermore, we were able to find the associated sensory and accessory (support) cells (*Figure 1C*). The spatial pattern and abundance of types of sensilla differ between segments. We noticed varying arrangements for T1, T2-T3, and A1-A7, with a consistent sequence of sensilla in each configuration. The head and the last abdominal segments represent unique cases with specialized sense organs like the terminal sensory cones (t) (*Singh and Singh, 1984*; *Dambly-Chaudière and Ghysen, 1986*).

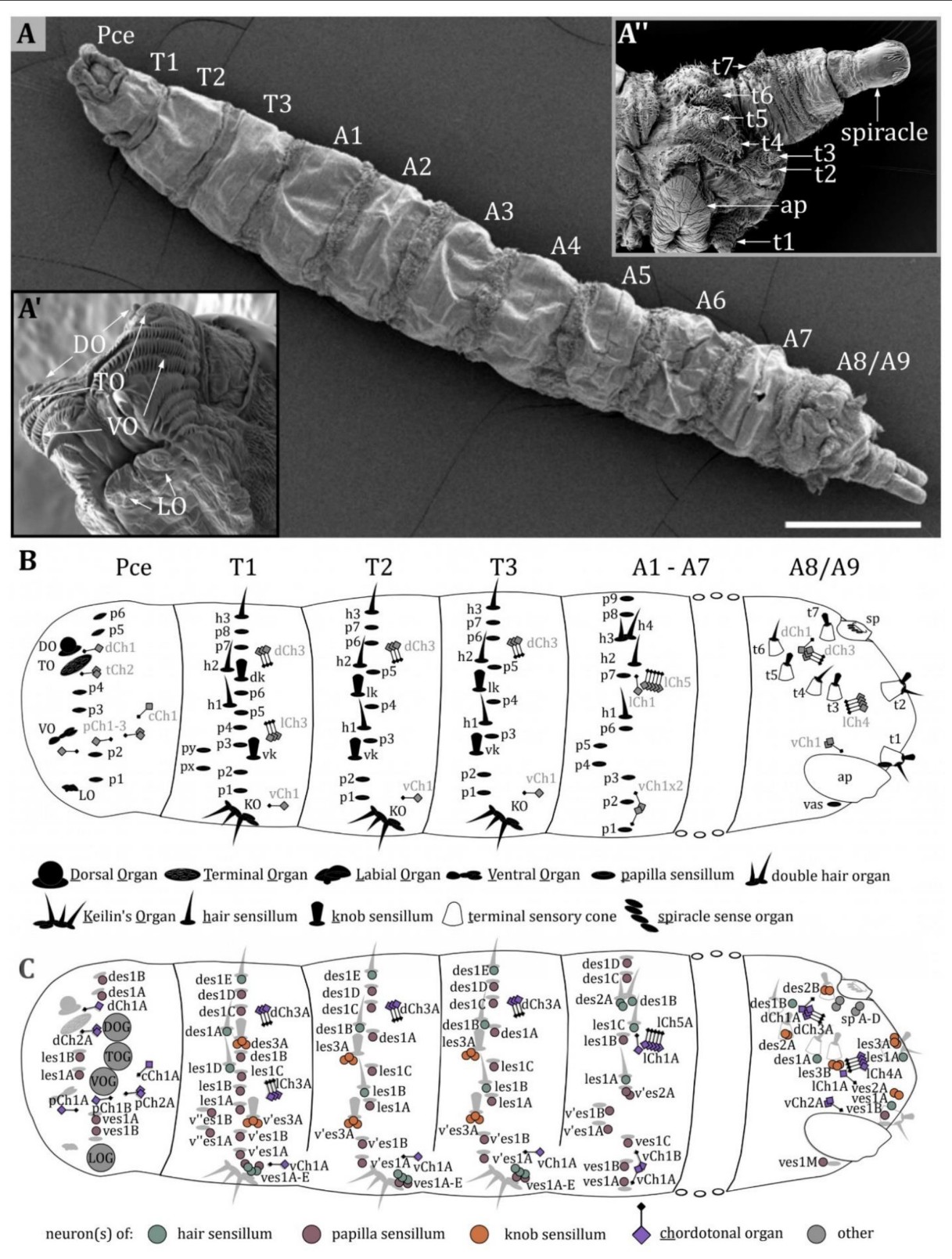

**Figure 1.** Larval body organization and a spatial map of external and chordotonal sense organs. (**A**) Ventral view of an third instar (L3) *Drosophila* larva. The larval body is divided into the pseudocephalon (pce), three thoracic (T1–T3), and nine abdominal (A1–A9) segments. (**A'**) Latero-frontal view of the pce. Major sensory organs of the head are bilaterally organized and consist of several sensilla: the dorsal organ (DO) and terminal organ (TO) on the tip of the head lobes; the ventral organ (VO) located between rows of cuticle hairs; the labial organ (LO) below the mouth opening. (**A''**) Lateral view of

*Figure 1 continued on next page*

*Figure 1 continued*

the last fused abdominal segments A8 and A9, called the anal division. Sensilla in this region are organized in terminal sensory cones (t1–t7). On the ventral side, the anal plate (ap) is visible; towards the posterior end, the posterior spiracles are located. (**B**) Map of sensilla on pseudocephalon, thoracic, and abdominal segments. We categorized types of sensilla as papilla (p)-, knob (k)- and hair (h) sensilla. The major head organs, the keilin's organ (KO), the terminal sensory cones, and the spiracle sense organ (sp) are formed by several sensilla. The organization of sensilla is bilateral. Sensilla have fixed positions on hemisegments; therefore, all segments exhibit a fixed sensillar pattern. We noticed varying arrangements for T1, T2-T3, and A1-A7, with a consistent sequence of sensilla in each configuration. The pseudocephalon and the anal division display a structural organization very different from the other segments. The schematic summarizes the result of our investigation and includes results from previous studies (*Dambly-Chaudière and Ghysen, 1986*; *Green and Hartenstein, 1997*). (**C**) Map of neurons innervating the external sense organs and the chordotonal organs. Again, the results and names from previous studies (*Dambly-Chaudière and Ghysen, 1986*; *Green and Hartenstein, 1997*) were used but slightly modified to create a standardized nomenclature (see discussion). Scale bars: (**A**) 500 μm. Abbreviations: TOG – terminal organ ganglion; DOG – dorsal organ ganglion; VOG – ventral organ ganglion; LOG – labial organ ganglion. For the nomenclature of neurons in (**C**), see discussion.

Papilla sensilla are most abundant in terms of number. We counted six papilla sensilla on the head, ten papilla sensilla on T1, seven on T2-T3, and nine on A1-A7 (*Figures 1 and 10*). Additionally, we find one papilla sensillum in the terminal sensory cone $t_1$ of segment A8/A9. Please note that the numbers are given per hemisegment, which is true for all sensilla but the unpaired ventral papilla sensillum (vas). We found three hair sensilla on T1-T3 and four hair sensilla on A1-A7, including the two hair sensilla $h_3$ and $h_4$ of the double hair organ (*Figure 1B*, *Figures 11 and 12*). Four hair sensilla are organized in the terminal sensory cones $t_1$, $t_2$, $t_4$, and $t_6$ in segments A8/A9. The KO is exclusively found on thoracic segments and is situated on the ventral side of each hemisegment of T1 -T3 (*Figure 1B* and *Figures 14 and 15*). Knob sensilla are found on the thoracic segments but also in the TO of the head and in the terminal sensory cones, although the number of neurons innervating these knob sensilla differs from one to three (*Figures 1, 11 and 16*). Two knob sensilla are restricted to the TO (*Figure 16*) and T1 - T3 (*Figure 1B* and *Figure 13*). In A8/A9 we find five knob sensilla in the terminal sensory cones $t_1$, $t_2$, $t_3$, $t_5$, and $t_7$ (*Figure 16G*). In addition, we find the spiracle sense organ in the last segment, which consists of four papilla-like sensilla located at the posterior spiracle (*Figure 17*). This sensilla configuration is based on the single ssTEM and various SEM scans we compared with published data (*Lohs-Schardin et al., 1979*; *Dambly-Chaudière and Ghysen, 1986*; *Green and Hartenstein, 1997*).

In the following, sense organs and sensilla of the larval head, thoracic, and abdominal segments are classified based on their external and internal morphology. Also, developmental aspects are addressed by comparison of first (L1) and third instar (L3) larvae.

## Description of individual external sensory organs of the *Drosophila* larva

### Dorsal organ

Despite previous studies on the ultrastructure of the DO (*Chu and Axtell, 1971*; *Singh and Singh, 1984*), many aspects of its structural organization, like the peripheral sensilla and the corresponding accessory cells (ACs), remained unclear. The 3D EM volumes recorded in the present work allow to complement and clarify the knowledge of the anatomy of the DO (*Figures 2–7*, *Table 1*). *Figures 2–7* provide a comprehensive description of the organization of the entire organ. Therefore, by means of targeted FIB-SEM and ssTEM, we obtained continuous image stacks of the DO covering the distance from its outer sensory parts to the region of the DO ganglion (DOG), where the cell bodies of the sensory neurons are located. It enabled us to trace dendrites from their tips at the cuticle surface to the neuron's cell body in the DOG and to generate a 3D reconstruction visualizing the spatial organization of the neuronal components of the DO (*Figure 2A and C*). Furthermore, accessory support cells were identified and described in detail (*Figure 7*). Most sensilla display a repertoire of these cells in a highly stereotyped fashion: a thecogen, a trichogen, and a tormogen cell. Due to their role in the formation of the sensillum, they have also been termed sheath, shaft, and socket cell, respectively (*Prelic et al., 2021*).

The DO is the primary olfactory organ of the larva. Its central structure is the dome (*Figure 2B–E*), which comprises seven olfactory sensilla with three dendrites each (*Figure 3B–F*), as reported in the literature (*Chu and Axtell, 1971*; *Singh and Singh, 1984*). Olfactory dendrites branch multifold in the dome region of the DO and connect to pore tubules spanning the outer cuticle, each leading to one of the tiny pores perforating the dome (*Figure 2B, D and E*). Apart from these tiny pores, the dome

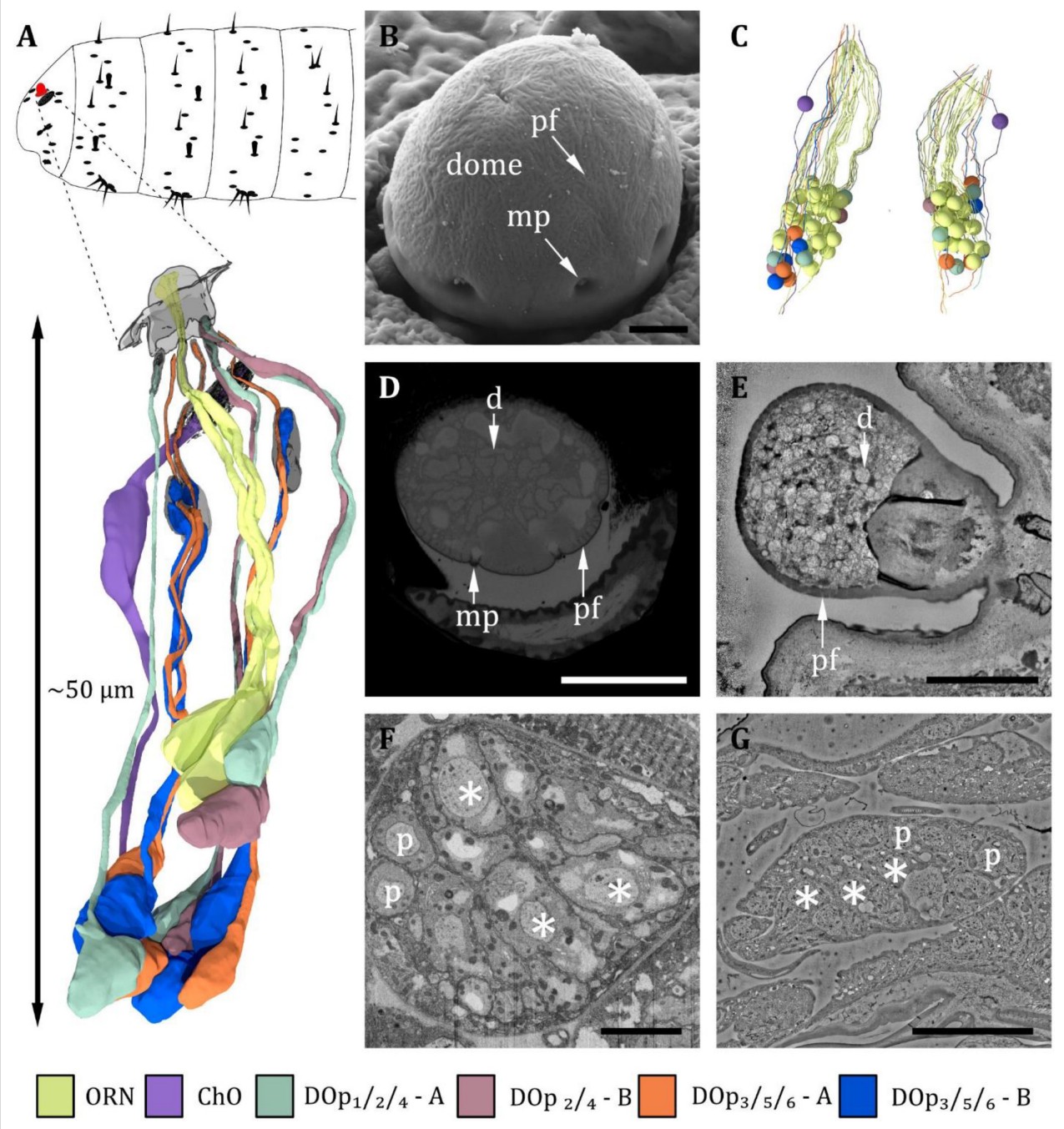

| ORN | ChO | DOp$_{1/2/4}$ - A | DOp$_{2/4}$ - B | DOp$_{3/5/6}$ - A | DOp$_{3/5/6}$ - B |
|---|---|---|---|---|---|

**Figure 2.** Ultrastructure of the dorsal organ – an overview. (**A**) 3D reconstruction of a representative dorsal organ of a first instar (L1) larva highlighting its position at the larval body (top) and its cell type organization. Color scheme: yellow- olfactory receptor neuron (ORN); –purple – chordotonal organ (ChO); blue – DO$_{p3/5/6}$ cooling cell; orange – DO$_{p3/5/6}$ warming cell; turquoise – DO$_{p1/2/4}$ mechanosensory cell; red – DO$_{p2/4}$ unknown sensory cell. Color code in (**A**) applies to all micrographs in *Figures 2–6* (**B**) SEM image depicting the outer morphology of the dorsal organ in a third instar (L3) larva. The dome is covered by multiple tiny perforations (pf). Three of the seven molting pores, which are traces of ecdysis, are visible. (**C**) Reconstruction of the peripheral and olfactory sensilla of the dorsal organ (DO) from their sensory tip to their cell body in the dorsal organ ganglion (DOG). (**D**) L3: Cross-section of the dome with tiny perforations and molting pores. The olfactory triplets are still visible, but dendritic branching is already present at this level. (**E**) L1: longitudinal section of the dome with tiny perforations but absent molting pores, as ecdysis has not occurred yet. The olfactory dendrites are branched and spread throughout the dome. (**F**) L3: ganglion of the dorsal organ, showing exemplary peripheral (p) and olfactory (asterisks) cell bodies (**G**) L1: ganglion of the dorsal organ, showing exemplary peripheral (p) and olfactory (asterisks) cell bodies. Scale bars: (**B**) 2 µm; (**D**) 5 µm; (**E**) 2 µm; (**F**) 5 µm; (**G**) 10 µm. Abbreviations: d - dendritic branches; mp - molting pores; p –peripheral cell bodies; pf – perforations.

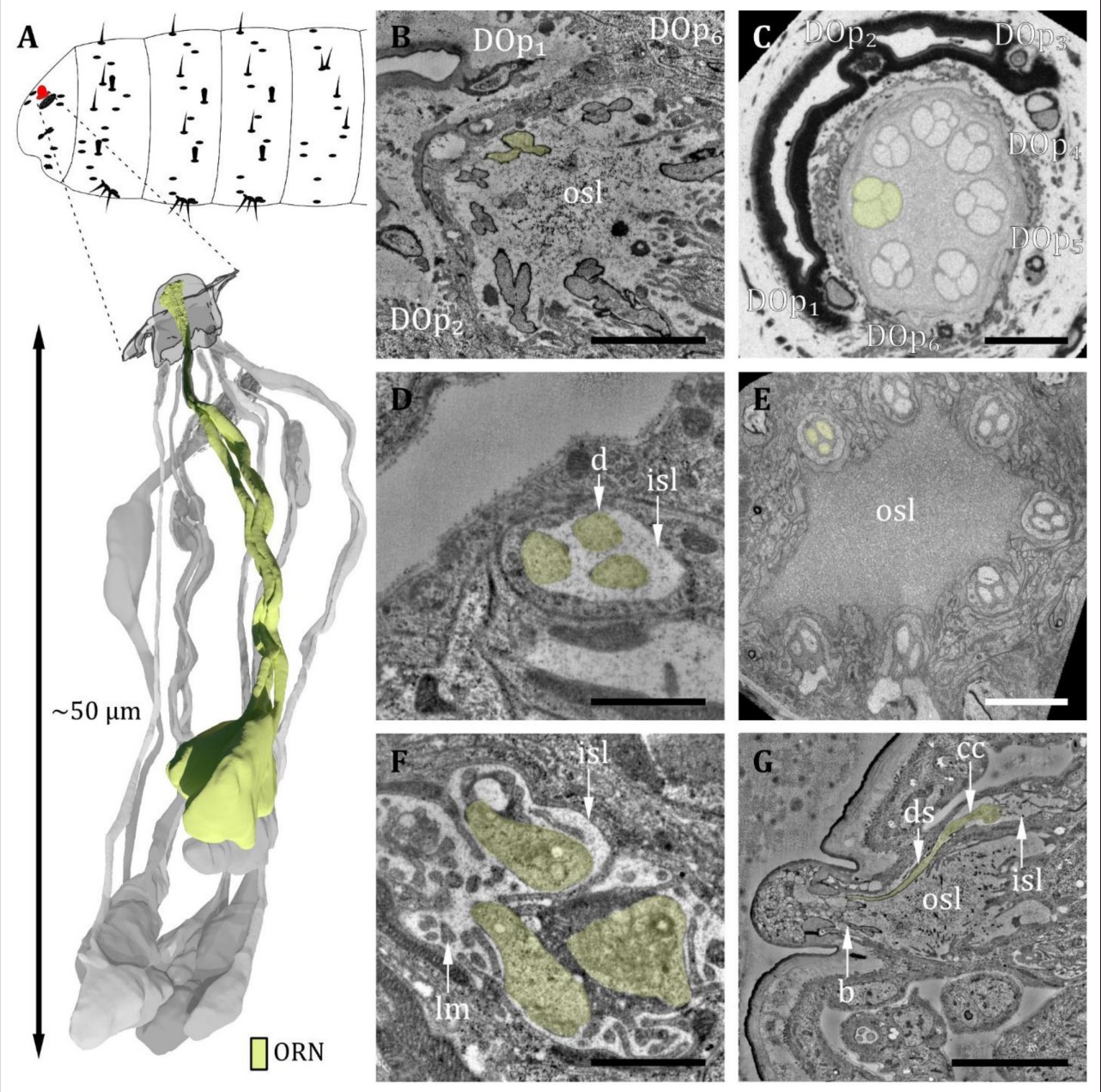

**Figure 3.** Ultrastructure of the dorsal organ – olfactory sensilla. (**A**) 3D reconstruction of a dorsal organ of a first instar (L1) larva showing its position at the larval body (top) and one exemplary triplet of olfactory receptor neurons (ORNs) (yellow, bottom). The color code in A applies to all micrographs in this figure. (**B**) L1: A longitudinal section through the base of the dorsal organ showing seven olfactory triplets bathed in the outer sensillum lymph (osl). Peripheral sensilla (DOp) 1, 2, and 6 are visible (**C**) L3: A cross-section through the base of the dorsal organ showing seven olfactory triplets and all peripheral sensilla 1–6. (**D**) L1: olfactory triplet at the level of the inner sensillum lymph (isl) cavity. (**E**) L3: olfactory triplets further proximal than in C. (**F**) L1: dendritic inner segments of the olfactory triplet bathed in the isl. The thecogen support cell is highly lamellated (lm). (**G**) L1: longitudinal section through an olfactory sensillum from dendritic branching (**b**) in the dome through osl where the dendrite is enclosed by a dendritic sheath (ds) to the ciliary constriction (cc) at the transition from the outer to the inner dendritic segment inside the isl cavity. Scale bars: (**B**) 2 µm; (**C**) 5 µm; (**D**) 1 µm; (**E**) 5 µm; (**F**) 1 µm; (**G**) 5 µm. Abbreviations: $DO_{p1-p6}$ – peripheral sensilla 1–6; b – (dendritic) branching; cc - ciliary constriction; d - dendrite; ds - dendritic sheath; inner sensillum lymph – isl; lc - lymph cavity; lm – lamellation; osl – outer sensillum lymph; ORN – olfactory receptor neuron.

The online version of this article includes the following figure supplement(s) for figure 3:

**Figure supplement 1.** Scanning electron microscopy (SEM) image of the larval head seen from above.

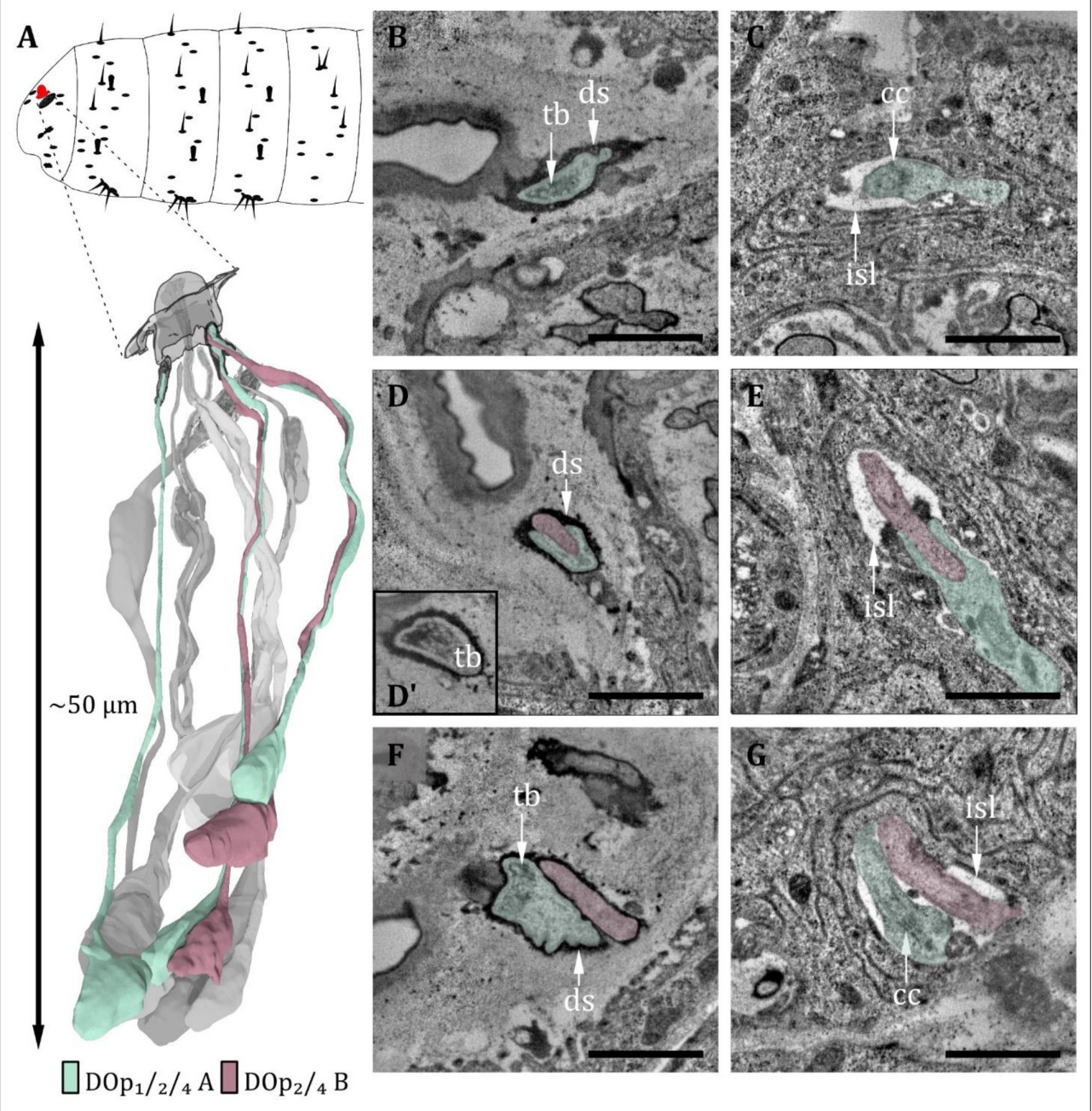

**Figure 4.** Ultrastructure of the dorsal organ – peripheral sensilla 1, 2, and 4. (**A**) 3D reconstruction of a dorsal organ of a first instar (L1) larva showing its position at the larval body (top) and the peripheral sensilla $DO_{p1/2/4}$ and their innervating neurons $DO_{p1/2/4-A}$ and $DO_{p2/4-B}$ (bottom). The color code in A applies to all micrographs in this figure. (**B**) DOp1 at the sensory tip with the sensory dendrite Dop1A containing a tubular body (tb). The dendrite is enclosed by a dendritic sheath (ds). (**C**) $DO_{p1A}$ at the level of the ciliary constriction (cc) inside the inner sensillum lymph (isl) cavity. (**D**) $DO_{p2}$ at the sensory tip with ds, the dendrite of $DO_{p2B}$ does not contain a tb in contrast to $DO_{p2A}$. (**D′**) $DO_{p2A}$ at the sensory tip with tb. (**E**) $DO_{p2A}$ and $DO_{p2B}$ at the level of the cc inside the isl cavity. (**F**) $DO_{p4}$ at the sensory tip with ds. $DO_{p4A}$ contains a tb, whereas $DO_{p4B}$ does not. (**G**) $DO_{p4A}$ and $DO_{p4B}$ at the level of cc bathed in the isl. Scale bars: (**B**) 1 µm; (**C**) 1 µm (**D**) 1 µm; (**E**) 1 µm; (**F**) 1 µm; (**G**) 1 µm. Abbreviations: tb - tubular body; ds - dendritic sheath; cc - ciliary construction; isl - inner sensillum lymph.

is interspersed with seven larger pores arranged in a circle (*Figure 2B and D*). In contrast to L3, these pores are missing in L1 larvae (*Figure 2E*), which suggests that these are molting pores formed during ecdysis when olfactory sensory neurons pass through the developing cuticle of the new DO. The seven olfactory triplets are housed in a common outer sensillum lymph space, which is built by the ACs (*Figure 3B, C and E* and *Figure 7*). In the outer lymph space, the triplets are enclosed by a dendritic

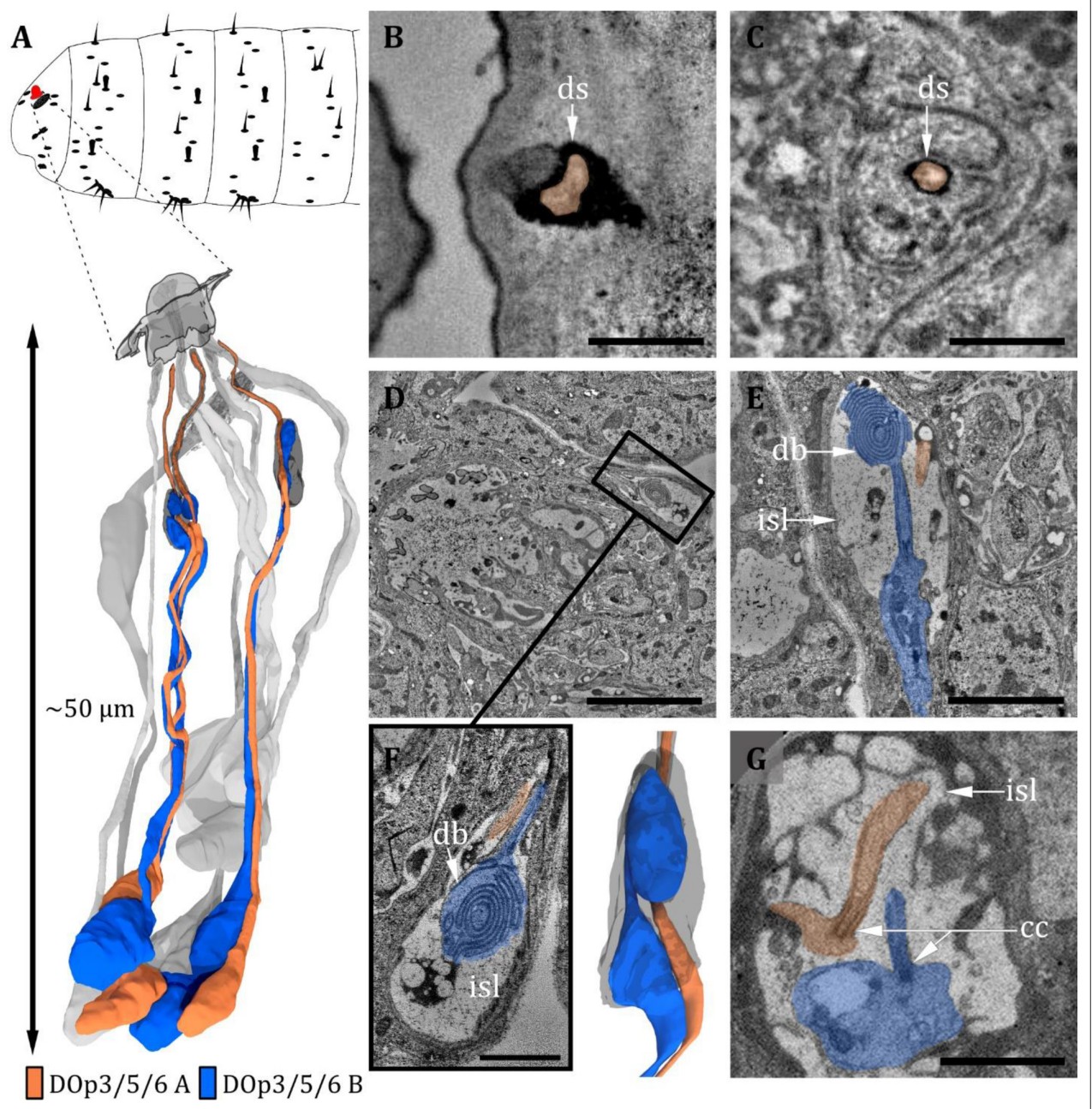

**Figure 5.** Ultrastructure of the dorsal organ – peripheral sensilla 3, 5, and 6. (**A**) 3D reconstruction of a dorsal organ of a first instar (L1) larva showing its position at the larval body (top) and the peripheral sensilla DO$_{p3/5/6}$, which exhibit the same morphology (bottom). DO$_{p3/5/6}$ are innervated by their corresponding neurons DO$_{p3/5/6-A}$ and DO$_{p3/5/6-B}$. The color code in A applies to all micrographs in this figure. (**B**) DO$_{p3B}$ is enclosed by a dendritic sheath (ds), which is connected to the epicuticle. (**C**) DO$_{p3}$ further proximal with DO$_{p3A}$ enclosed by a dendritic sheath and a thecogen cell. (**D–G**) DO$_{p3A}$ enters the inner sensillum lymph (isl) cavity formed by the thecogen cell; the ds is no longer visible. The dendrite is transforming from the outer to the inner dendritic segment through a ciliary constriction (cc). The same is true for the dendrite of DO$_{p3B}$, but it is not projecting through the ds towards the cuticle. Instead, it forms a lamellated bulbous structure within the lymph cavity, the dendritic bulb (db). (**D**) Longitudinal section of peripheral sensillum DO$_{p5}$ at the level of the isl cavity, which is formed by thecogen cell. (**E**) Longitudinal section of DO$_{p3}$ at the level of the isl cavity. DO$_{p3B}$ enters the cavity proximally, where it transforms from an enlarged inner dendritic segment (ids) to a ciliary outer dendritic segment (ods). The cilium forms a lamellated db inside the cavity. DO$_{p3A}$ appears rather inconspicuously but protrudes towards the cuticle in contrast to DO$_{p3B}$. (**F**) Close-up view of the isl cavity in (**D**), the db of DO$_{p5B}$ is visible. DO$_{p5A}$ is projecting inside the ds towards the cuticle. (**G**) L3: peripheral sensillum DO$_{p3}$ at the ciliary constriction. Both ids are heavily swollen. Scale bars: (**B**) 0.5 µm; (**C**) 0.5 µm; (**D**) 5 µm; (**E**) 2 µm; (**F**) 1 µm; (**G**) 2 µm. Abbreviations: db - dendritic bulb; ds - dendritic sheath; cc - ciliary constriction; isl - inner sensillum lymph.

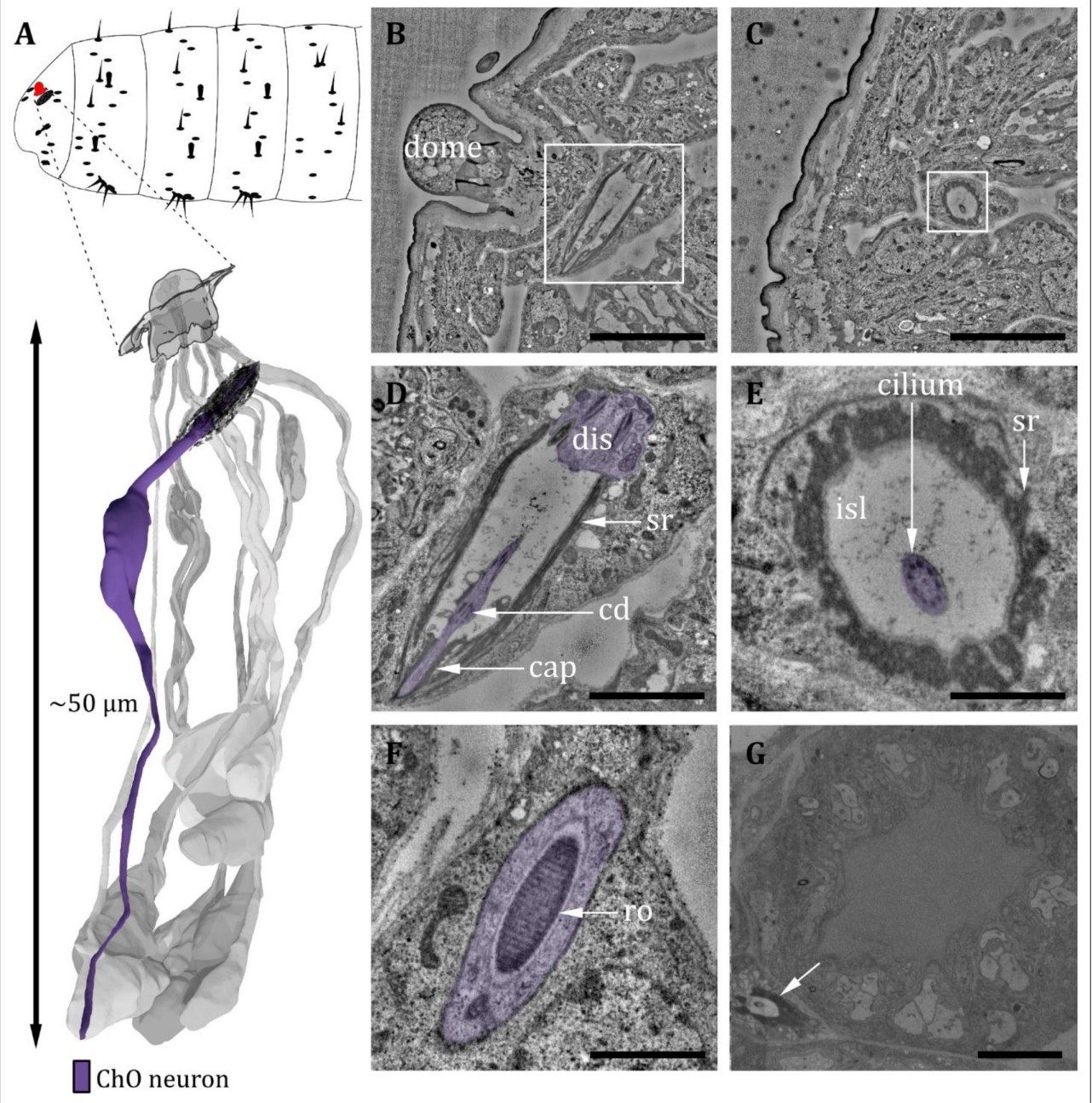

**Figure 6.** Ultrastructure of the dorsal organ – the chordotonal organ. (**A**) 3D reconstruction of a dorsal organ of a first instar (L1) larva showing the position of the chordotonal organ dCh$_{1A}$ at the larval body (top) and its cellular organization (bottom). The color code in A applies to all micrographs in this figure. (**B**) Longitudinal section of dCh$_{1A}$ at left DO (white box) within the whole larval volume. (**C**) Cross-section of dCh$_{1A}$ at right DO (white box) within the whole larval volume. (**D**) Presentation of the white box shown in B at higher magnification. The dendrite (purple) is inserted into the scolopale made of the scolopale rods (sr). The dendritic inner segment (dis) and the cilium with a ciliary dilation (cd) are visible. The cilium is inserted into a cap at the distal end. (**E**) Presentation of the white box shown in C at higher magnification. The sr encloses the inner sensillum lymph (isl), in which the cilium is bathed. (**F**) Dis of dCh$_{1A}$ with striated ciliary rootlet (ro). (**G**) The dCh$_{1A}$ is also present in L3 larvae (white arrow). Scale bars: (**B**) 5 µm; (**C**) 5 µm; (**D**) 2 µm; (**E**) 1 µm; (**F**) 1 µm; (**G**) 5 µm. Abbreviations: cd – ciliary dilation; ChO – chordotonal organ; dis – dendritic inner segment; isl - inner sensillum lymph; ro - ciliary rootlet; sr - scolopale rods.

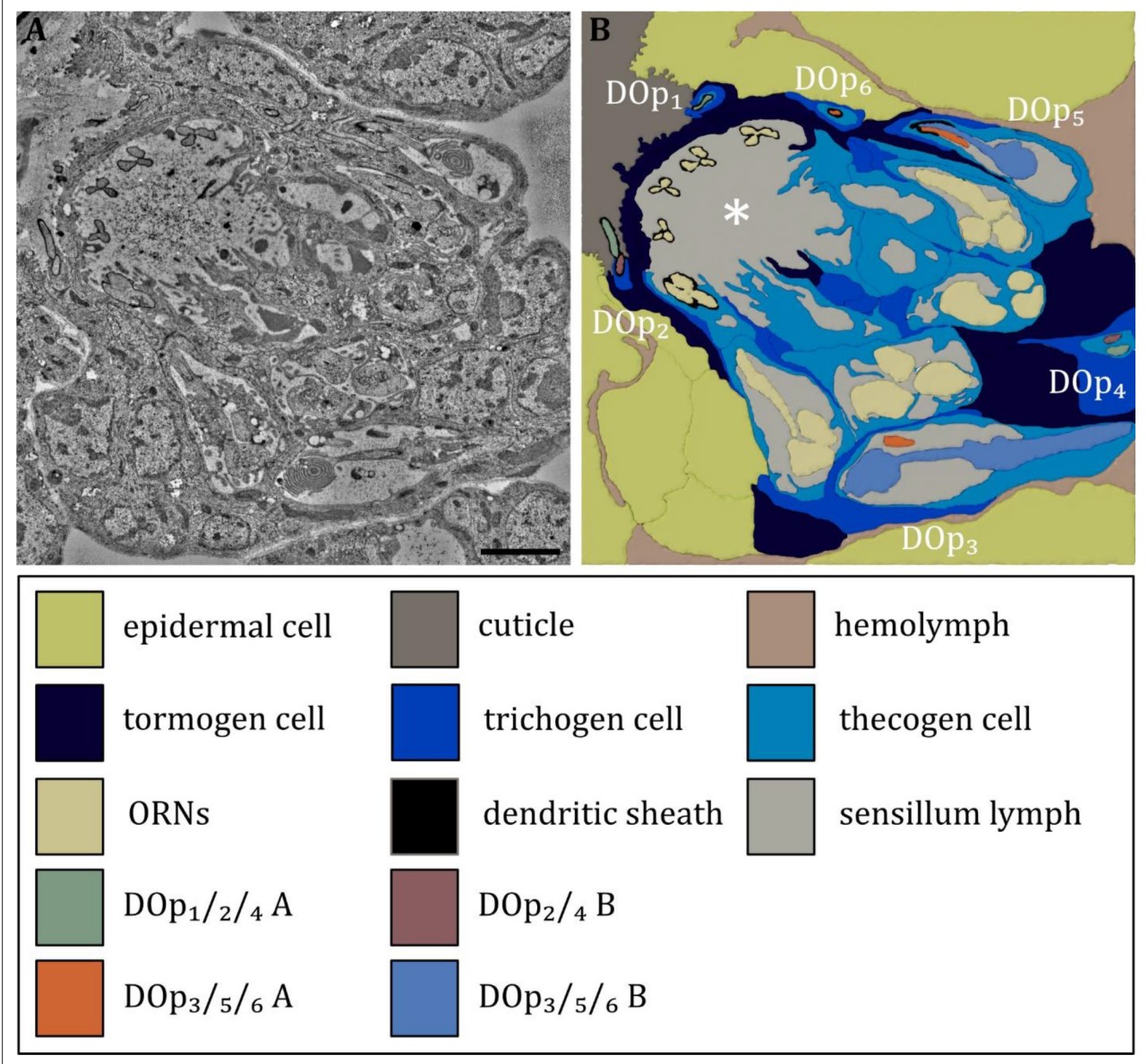

**Figure 7.** Ultrastructure of the dorsal organ – the sensilla and associated support cells. (**A**) Overview of the dorsal organ proximal to the sensory dome (to the upper left corner). (**B**) Schematic drawing of (A) defining all olfactory and peripheral sensilla and their associated support cells. All sensilla possess their individual set of support cells. The thecogen cells form an inner sensillum lymph space for each individual sensillum. The trichogen and tormogen cells wrap around the thecogen cell and contribute to the sensillar integrity. In addition, the olfactory support cells build up a common outer sensillum lymph space (asterisk). Abbreviations: DO$_{p1-p6}$ – dorsal organ peripheral sensilla 1–6; ORNs – olfactory receptor neurons; Scale bar: 2 µm.

sheath (*Figure 3G*), which is most likely segregated by the thecogen cell (*Figure 7*). Further distal, the three olfactory dendrites are bathed in a small inner sensillum lymph cavity enclosed by the thecogen cell (*Figure 3D and F*), also called the perineuronal lumen (*Prelic et al., 2021*). Inside the cavity lays the ciliary constriction (cc), which marks the transition from the outer dendritic segment (ods) to the inner dendritic segment (ids) (*Figure 3G*).

All cell bodies of DO sensory neurons lie in the dorsal organ ganglion (DOG), in which the soma of olfactory receptor neurons (ORNs) are located more to the center compared to those of the peripheral sensilla (*Figure 2A and C*). The cell bodies of neurons innervating individual peripheral sensilla lie adjacent (*Figure 2A and C*). Cell bodies of olfactory neurons appear elliptically shaped and are relatively large compared to peripheral neurons' cell bodies, which have a roundish appearance and are smaller in diameter (*Figure 2F and G*). Also located within the DOG are additional cell bodies of

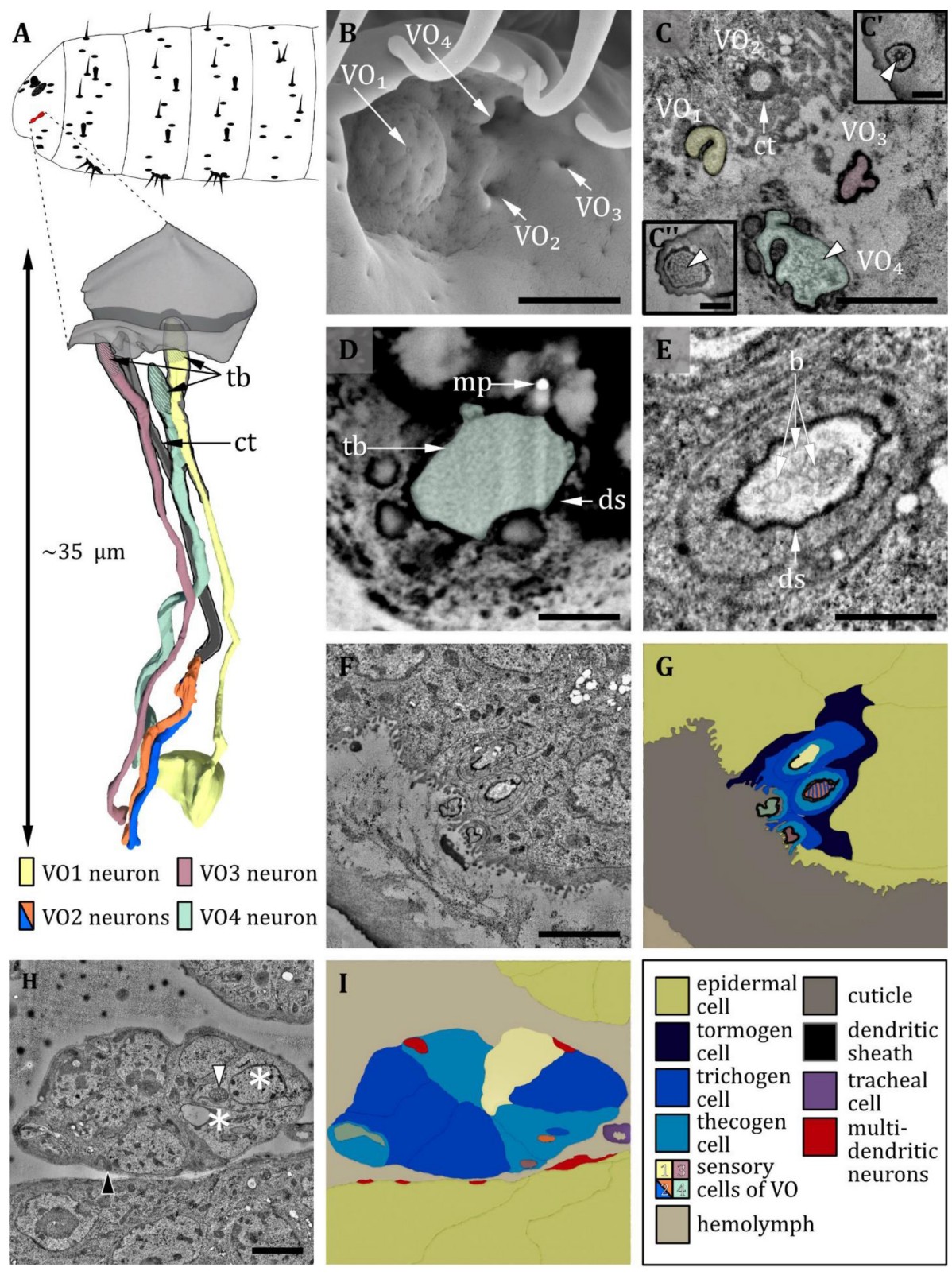

**Figure 8.** Ultrastructure of the ventral organ. (**A**) 3D reconstruction of the ventral organ of an L1 larva (bottom). The outline shows the position of this organ on the larval head (top). (**B, D, E**) (FIB)-SEM images of L3 larvae; (**C, F, G, H**) section scanning transmission electron microscopy (ssTEM) images of first instar (L1) larvae. The color code in A applies to all micrographs in this figure. (**B**) Electron micrograph of the ventral organ in third instar (L3) larva consisting of four sensilla, which were named $VO_1$-$VO_4$. $VO_1$ is arched outwards like a dome, whereas $VO_{2-4}$ lies in small depressions. (**C**) Cross-section

*Figure 8 continued on next page*

*Figure 8 continued*

of the ventral organ in an L1 larva. VO$_{1/3/4}$ possess a tubular body each (tb; white arrowheads). VO$_2$ consists of a cuticle tube (ct) with a terminal pore. (**C′**) tubular body of VO$_3$. (**C″**) tubular body of VO$_1$. (**D**) Cross-section through the base of VO$_4$ in an L3 larva. The dendrite with tb and dendritic sheath (ds) is anchored in the cuticle by electron-dense material. A putative molting pore (mp) can be observed. (**E**) VO$_2$ in an L3 larva. The pore channel is streaked by fine dendritic branches (b). (**F**) Cross-section through the ventral organ further proximal than in (**C**). (**G**) Schematic drawing of (**F**), showing the associated sensory and support cells. (**H, I**) Micrograph (**I**) and schematic drawing (**H**) of the ventral organ further proximal than (**F**) at the level of the ganglion, defining the sensory cells and the associated support cells. The two dendrites of VO$_2$ become clearly visible when the pore enters a greater cavity formed by the thecogen cell. The dendrites then penetrate the thecogen cell, which exhibits a particular appearance like the support cells of the pit sensilla of the terminal organ (not shown). Those cells appear to be electron-lucent (asterisks), and their mitochondria exhibit a tubular (white arrowhead) instead of a cristae type (black arrowhead). Scale bars: (**B**) 1 μm; (**C**) 1 μm, inlets 0.5 μm; (**D**) 0.5 μm; (**E**) 0.5 μm; (**F**) 1 μm; (**G**) 2 μm; (**H**) 2 μm. Abbreviations: b – dendritic branches; ds - dendritic sheath; ct - cuticle tube; mp – molting pore; tb - tubular body.

the sensory cells of the papillum sensillum of the dorsolateral group of the TO (Pdo), seven additional non-sensory cells of unknown function but most likely glial cells, and one sensory cell of the papilla sensillum (p6) in close proximity to the DO. Their cell bodies lie in the proximal end of the DOG from where they wrap around the whole ganglion.

The six peripheral sensilla lie in the rim between the dome and the surrounding cuticle ridge. They are not visible by external investigation (*Figure 2B* and *Figure 3B and C*). The outer cuticle part of all six peripheral sensilla appears similar in its structure. Its organization is very simple, consisting of a small cuticle bulge with a tiny pore in its center (*Figure 3C*). The pore is first surrounded by a short cuticle channel that leads to the dendrites (*Figure 3C*). Remarkably, the pores are absent in L1 larvae (*Figure 3B*) and are, therefore, most likely molting pores. Internally, however, peripheral sensilla have different structural properties (*Figure 4* and *Figure 5*).

The spatial arrangement of the peripheral sensilla was consistent in all analyzed samples. This stereotypical pattern allowed us to number the peripheral sensilla DO$_{p1}$ to DO$_{p6}$ (dorsal organ peripheral sensilla 1–6). DO$_{p1}$ is the posterior most sensillum. The six DO$_p$ were numbered from 1 to 6 in a clockwise direction in the left and anti-clockwise direction in the right hemisphere when seen from above (see *Figure 3—figure supplement 1*).

Within these six sensilla, we find three structurally similar types (type 1: DO$_{p1}$; type 2: DO$_{p2}$ and DO$_{p4}$; type 3: DO$_{p3}$, DO$_{p5}$, and DO$_{p6}$; see *Table 1*). DO$_{p1}$ is characterized by a single dendrite with a tubular body that terminates at the base of the epicuticle (*Figure 4B and C*). DO$_{p2}$ and DO$_{p4}$ can be identified by the presence of two dendrites; one (DO$_{p2A}$ and DO$_{p4A}$) ends with a tubular body, the other one (DO$_{p2B}$ and DO$_{p4B}$) not (*Figure 4D–G*). Both dendrites are enclosed by a common dendritic sheath. Similar to the olfactory neurons, DO$_{p1/2/4}$ dendrites are bathed in a small inner sensillum lymph space which is enclosed by the thecogen cell (*Figure 4C, E and G*). DO$_{p3}$, DO$_{p5}$, and DO$_{p6}$ share a similar structural organization and house two dendrites each (*Figure 5F and G*). DO$_{p3A}$, DO$_{p5A}$, and DO$_{p6A}$ end below the epicuticle and are enclosed by a dendritic sheath (*Figure 5B and C*). DO$_{p3B}$, DO$_{p5B}$, and DO$_{p6B}$ end inside the inner sensillum lymph space.

The dendrites of DO$_{p3}$, DO$_{p5}$, and DO$_{p6}$ form the ciliary constriction inside an inner sensillum lymph cavity, like other DO sensilla. DO$_{p3/5/6-A}$ has a small dendritic outer and a larger dendritic inner segment like canonical receptor neurons. In contrast, DO$_{p3/5/6-B}$ are lamellated and form membrane staples, called dendritic bulbs (db), inside the lumen (*Figure 5D–G*). In L1 larvae, their dendrite terminates clearly inside the lumen; in L3 larvae, it might extend further into the dendritic sheath. Besides the dendritic bulb, the dendritic inner segment is heavily swollen after the ciliary constriction, forming another bulb-like structure (*Figure 5E–G*). This fits with confocal image data, where the dendritic bulb seems to be divided (*Ni et al., 2016*). In the DOG, the cell bodies of both neurons lie adjacent (*Figure 2A and C*).

In addition to the peripheral sensilla DO$_{p1-6}$, we find one papilla sensillum, p$_6$, in close proximity to the DO, whose sensory cell body (des$_{1B}$) lies in the DOG. The structure of papilla sensilla will be presented in a separate results section (see *Figure 10*).

It is difficult to compare our findings to *Singh and Singh, 1984* because their description of peripheral sensilla is vague. In *Musca*, *Chu and Axtell, 1971* classified the peripheral sensilla into four types: contact chemoreceptor, unclassified receptor, lateral pore receptor, and scolopidium-like receptor. Based on the number of dendrites and other structural properties, DO$_{p4}$ and DO$_{p2}$ might correspond to the unclassified receptor, and DO$_{p5}$ and DO$_{p3}$ might correspond to the contact chemosensory receptor. DO$_{p1}$ most likely corresponds to the scolopidium-like receptor, even though the classification

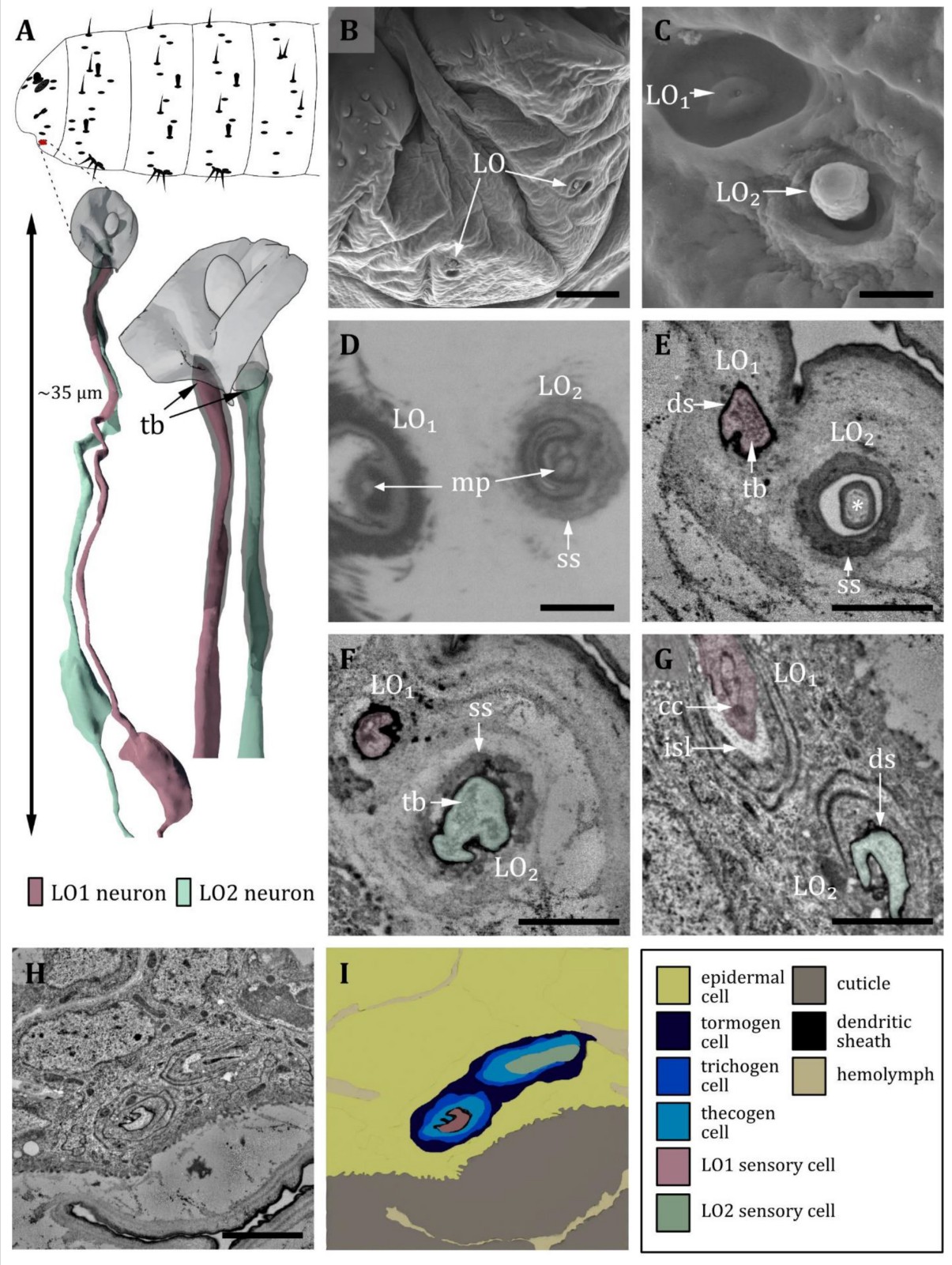

**Figure 9.** Ultrastructure of the labial organ. (**A**) 3D reconstruction of the labial organ of a first instar (L1) larva (bottom). The outline shows the position of this organ on the larval head (top). (**B, C, D**) (FIB)-SEM images of third instar (L3) larvae; (**E, F, G, H**) section scanning transmission electron microscopy (ssTEM) images of L1 larvae. The color code in A applies to all micrographs in this figure. (**B**) Scanning electron microscopy (SEM) image of the lower larval head with left and right labial organ (LO). (**C**) SEM image of the LO in an L3 larva. The LO is composed of two sensilla, here named LO₁ and LO₂.

*Figure 9 continued on next page*

Figure 9 continued

LO1 forms a pore in the center of a small socket on the bottom of a cylindrical cuticle depression. $LO_2$ protrudes peg-shaped from the cuticle; its outer cuticle structure appears rough. (**D**) Cross-section through the base of $LO_1$ and $LO_2$. Here, putative molting pores (mp) are visible. $LO_2$ is enclosed by a socket septum (ss). (**E**) Cross-section of the base of $LO_1$ and $LO_2$. $LO_1$ contains a tubular body and is enclosed by a dendritic sheath. The knob of $LO_2$ is mainly formed by the epicuticle and partially by the exocuticle at the base (asterisk). (**F**) Cross-section proximal of (**E**). $LO_1$ and $LO_2$ are enclosed by a dendritic sheath. In $LO_2$, the tubular body is visible. The whole knob-shaped structure is held in place by the socket septum. (**G**) Cross-section proximal of (**F**). Dendrite of $LO_1$ has entered the inner sensillum lymph (isl) cavity. The ciliary constriction (cc) is visible at the transition between the inner and outer segments. $LO_2$ is still enclosed by the dendritic sheath. (**H**) Cross-section through the labial organ showing the enveloping support cells. (**I**) Schematic drawing of (**H**): The dendrites are surrounded by their sensillar support cells, the thecogen (which forms the dendritic sheath), the trichogen, and the tormogen cell. Scale bars: (**B**) 10 µm; (**C**) 1 µm; (**D**) 1 µm; (**E**) 1 µm; (**F**) 1 µm; (**G**) 1 µm; (**H**) 2 µm. Abbreviations: cc - ciliary constriction; ds - dendritic sheath; isl – inner sensillum lymph; mp – molting pore; ss - socket septum; tb - tubular body.

is misleading, as scolopales are associated with chordotonal organs, which have no connection to the surface. From today's perspective, it would rather be classified as a campaniform or papilla sensillum, and misclassification might be due to the interpretation of dendritic sheaths as scolopales in campaniform sensilla.

The seven olfactory sensilla composing the dome of the DO share a similar structural organization (**Figure 3B–G**). Therefore, discrimination between them, like done for the peripheral sensilla, was not feasible. At the base of the dome, olfactory dendrites form seven tightly packed clusters of three dendrites (**Figure 3B and C**). These olfactory triplets lie in a circle immersed in common sensillum lymph space, which is filled with a substance of heterogeneous electron density (**Figure 3B and C**). The peripheral sensilla are arranged as a circle around the olfactory triplets. They lie outside of the sensillum lymph space (**Figure 3B, C and E**). Dendrites of all DO sensilla are enclosed by ACs further proximal to the dome base (**Figure 7**). The olfactory sheath cells form the common olfactory receptor lymph space. Further proximal, the olfactory triplets are bathed in electron-dense inner sensillum lymph inside their individual lymph cavity (**Figure 3D and F**, **Figure 7**). In comparison to dendrites of the peripheral sensilla, olfactory dendrites appear quite large in diameter, whereas $DO_{p3}$, $DO_{p5}$, and $DO_{p6}$ appear very tiny, especially in L3 larvae (**Figure 3C and E**).

The dendrites of DO sensilla are separated into an inner and an outer dendritic segment by a ciliary constriction inside the inner sensillum lymph cavity (**Figure 3G**; **4C, E, G**; **5E–G**). Dendritic inner segments of olfactory and peripheral sensilla are considerably larger in diameter than the outer segments, which is the typical structure of insect sensilla dendrites (**Keil, 1997**). A dendritic sheath surrounds the dendritic outer segments (e.g. **Figure 3G**; **4B, D, E**; **5B, C**) and disappears inside the lumen (**Figure 3G**). Dendrites of one sensillum seem to stay together after leaving their common lymph space and their cell bodies lie adjacent inside the DOG (**Figure 2A and C**).

In total, we find 43 cell bodies in the DOG, 36 of them being of sensory and seven being of non-sensory origin. The sensory cells in the DOG include the cell bodies of the 21 olfactory neurons, the 11 peripheral neurons, the neuron of the peripheral papilla sensillum, and the three neurons of the papillum $P_{do}$ of the dorso-lateral group of the TO (**Rist and Thum, 2017**). The seven non-sensory cells are of unknown origin, but most likely peripheral glial cells (see discussion).

Within the DO, we also find the (accessory) cells that build the structure of the organ. We find 60 cells in total, 28 (7 × 4) being associated with the olfactory sensilla, 24 (6 × 4) being associated with the six peripheral sensilla, and one being associated with the papilla sensillum $p_6$. The seven remaining cells are of unknown origin but might also be peripheral glial cells. The remaining ACs of $p_6$ and $P_{do}$ lay outside the DO.

## Chordotonal organ in close proximity to dorsal organ

In close proximity to the DO, we find a single-innervated (monodynal) chordotonal organ (ChO) (**Figure 6A and B**). It lays diagonally to the DO dome in between the non-sensory cells that build up the DO (**Figure 6C and D**), but it is not part of the DOG. The bipolar sensory cell ends with a ciliary structure of type 1 (**Yack, 2004**; **Figure 6E**). The cilium is growing out of the dendritic inner segment and is surrounded by the prominent scolopale, which is made up of the scolopale rods (**Figure 6D and E**). The rods are segregated by the scolopale cell. The end of the cilium is inserted into a cap, which is ensheathed by the cap cell (**Figure 6D**). The cilium is bathed in sensillum lymph within the scolopale (**Figure 6E**) and exhibits a ciliary dilation (**Figure 6D**). The inner dendritic segment contains

**Table 1.** Summary of morphological features of head sensory organs of third instar *Drosophila* larvae.

| Sensillum | External Structure | Pores | Dendrites | Hypothesized function |
|---|---|---|---|---|
| **DO** | | | | |
| dome | round, dome-shaped cuticle shaft | multiple pores | 21 arranged in seven triplets one with a tubular body | olfaction |
| $DO_{p1}$ | cuticle bulge with pore in center at dome-base | terminal pore (molting) | | mechanosensation |
| $DO_{p2}$, $DO_{p4}$ | cuticle bulge with pore in center at dome-base | terminal pore (molting) | two; one with a tubular body | |
| $DO_{p3}$, $DO_{p5}$, $DO_{p6}$ | cuticle bulge with pore in center at dome-base | terminal pore (molting) | two; 'dendritic bulbs' | mechanosensation + unknown thermosensation |
| **VO*** | | | | |
| $VO_1$ | shallow dome | terminal pore (molting) | one with a tubular body | mechanosensation |
| $VO_2$ | steep pore in cuticle surface | terminsal pore | two | contact chemosensation |
| $VO_3$, $VO_4$ | steep pore in cuticle surface | terminal pore (molting) | one with a tubular body | mechanosensation |
| **LO** | | | | |
| $LO_1$ | cuticle cavity with pore in center | terminal pore (molting) | one with a tubular body | mechanosensation |
| $LO_2$ | peg-shaped shaft, rough surface | basal pore (molting) | one with a tubular body | mechanosensation |
| **TO†** | | | | |
| $P_{1/3}$ | bud-shaped shaft with pore in the center | terminal pore | three; one with a tubular body | gustation + mechanosensation |
| $P_2$ | bud-shaped shaft with pore in the center | terminal pore | five; one with a tubular body | gustation + mechanosensation |
| $P_{do}$ | bud-shaped shaft with pore in the center | terminal pore | three | gustation |
| $P_{mod}$ | bud-shaped shaft | lateral pore (molting) | one with a tubular body | mechanosensation |
| $T_{1/5}$ | cuticle cavity with pore in center | terminal pore | three; one with a tubular body | gustation + mechanosensation |
| $T_2$ | cuticle cavity with pore in center | terminal pore | four | gustation |
| $T_{3/4}$ | cuticle cavity with pore in center | terminal pore | two | gustation |
| $S_{di/do}$ | pore in cuticle | terminal pore(molting) | one with a tubular body | mechanosensation |
| $K_1$ | knob-shaped shaft | lateral pore (molting) | one | chemosensation (oxygen) |
| $K_2$ | knob-shaped shaft | lateral pore (molting) | one (slightly lammelated) | osmosensation + chemosensation (oxygen) |

*__Singh and Singh, 1984__ report the VO to consist of five sensilla.
†adopted from __Rist and Thum, 2017__.

**Table 2.** Summary of morphological findings of this study for sensilla of thoracic and abdominal segments of first (L1) and third (L3) instar larvae.

| Sensillum | Number, location (*per hemisegment*) | External structure | Pores | Dendrites | Hypothesized function | Other names / classifications |
|---|---|---|---|---|---|---|
| hair sensillum (h) single double | total: 41 27 T1-3: 3 dorsal and lateral A1-7: 2 lateral A8: 4 at t1-2, t4, t6 7×2 hairs A1-7: 1 dorsal | hair-shaped smooth cuticle shaft, different sizes: short, intermediate, long; shaft might be bifurcated long, no bifurcation short and long; no bifurcation | L1: none L3: basal pore (molting) | **one**: terminates at base shaft with tb h3: **two**, one with and one w/o tb h4: **one** with tb | mechanosensation mechanosensation unknown function mechanosensation | hair-types C, D, E *Kankel, 1980* trichoid *Dambly-Chaudière and Ghysen, 1986*; *Green and Hartenstein, 1997*; *Hartenstein, 1988* |
| knob sensillum (k) | total: 11 T1: 2, one dorsal; one ventral T2-3: 2, one lateral, one ventral A8/9: 5 at t1-3, t5, t7 | knob-shaped cuticle shaft in cavity; smooth surface long; shaft protrudes out of cavity; smooth surface | L1: none L3: basal pore (molting) | T1-3: **three**: one innervates peg, one at base with tb, one at base w/o tb A8/9: **three** or **two** (one w/o tb is missing) | chemosensation (oxygen) mechanosensation unknown function | koelbchen *Dambly-Chaudière and Ghysen, 1986*; *Hertweck, 1931* knob-in-pit *Singh and Singh, 1984* hair-type B *Kankel, 1980* basiconic *Hartenstein, 1988* sensory papillae *Sato and Denell, 1985* dorsal/ ventral pit *Lewis, 1978* black sensory organ *Lohs-Schardin et al., 1979* black dot *Campos-Ortega and Hartenstein, 1985* |
| papilla sensillum (p) | total: 94,5 head: 6 T1: 10 T2-3: 7 A1-7: 9 dorsal, lateral, ventral A8: 1 at t1 +1 unpaired at anus (vas) | shallow cuticle depression (with pore in center in L3) | L1: none L3: terminal pore (molting) | **one**: one at base with tb **two**: p6 (A1-A7) one at base with tb, one w/o tb | mechanosensation mechanosensation unknown function | papilla *Dambly-Chaudière and Ghysen, 1986*; *Green and Hartenstein, 1997*; *Hartenstein, 1988* campaniform *Hartenstein, 1988*; *Singh and Singh, 1984* pit *Green and Hartenstein, 1997* |
| keilin's organ (KO) | total: 3 T1-3: 1 ventral | three hair sensilla: hair-shaped shaft of similar length two papilla sensilla: pore in cuticle surface in L3 | L1: none L3: basal pore (molting) L1: none L3: terminal pore (molting) | **five** (four in L3): one at base of each hair (3) one below each papilla (2) all with tb | mechanosensation mechanosensation | hair-type A *Kankel, 1980* Fussstummelsinnesorgan *Hertweck, 1931* |
| spiracle sense organ (sp) | total: 1 at posterior spiracle | four sensilla at base of the spiracular hairs | L1: none | **four**: one at base with tb | mechanosensation | spiracular hairs *Jürgens, 1987* hair tufts *Sato and Denell, 1985* |

a very noticeable striated ciliary rootlet (*Figure 6F*), which originates from the basal body towards the cell body. We find these DO-associated ChO (doChO) not only in L1 but also in L3 larvae (*Figure 6G*, white arrow).

## Ventral organ

The ventral organ is located ventral to the terminal organ and lateral to the mouth hooks. The sensilla of the VO are located in a cuticle invagination, hidden by a row of cirri (*Figure 8B*). We identified four sensilla in the VO (*Figure 8A–I*, *Table 1*). Three of them (VO$_1$, VO$_3$, and VO$_4$) are innervated by a single neuron, and one (VO$_2$) by two neurons (*Figure 8F–I*). This number is in accordance with previous findings in other cyclorrhaphan larvae (*Honda and Ishikawa, 1987*) but contradicts a study on *Drosophila* larvae reporting that five sensilla belong to the VO (*Singh and Singh, 1984*).

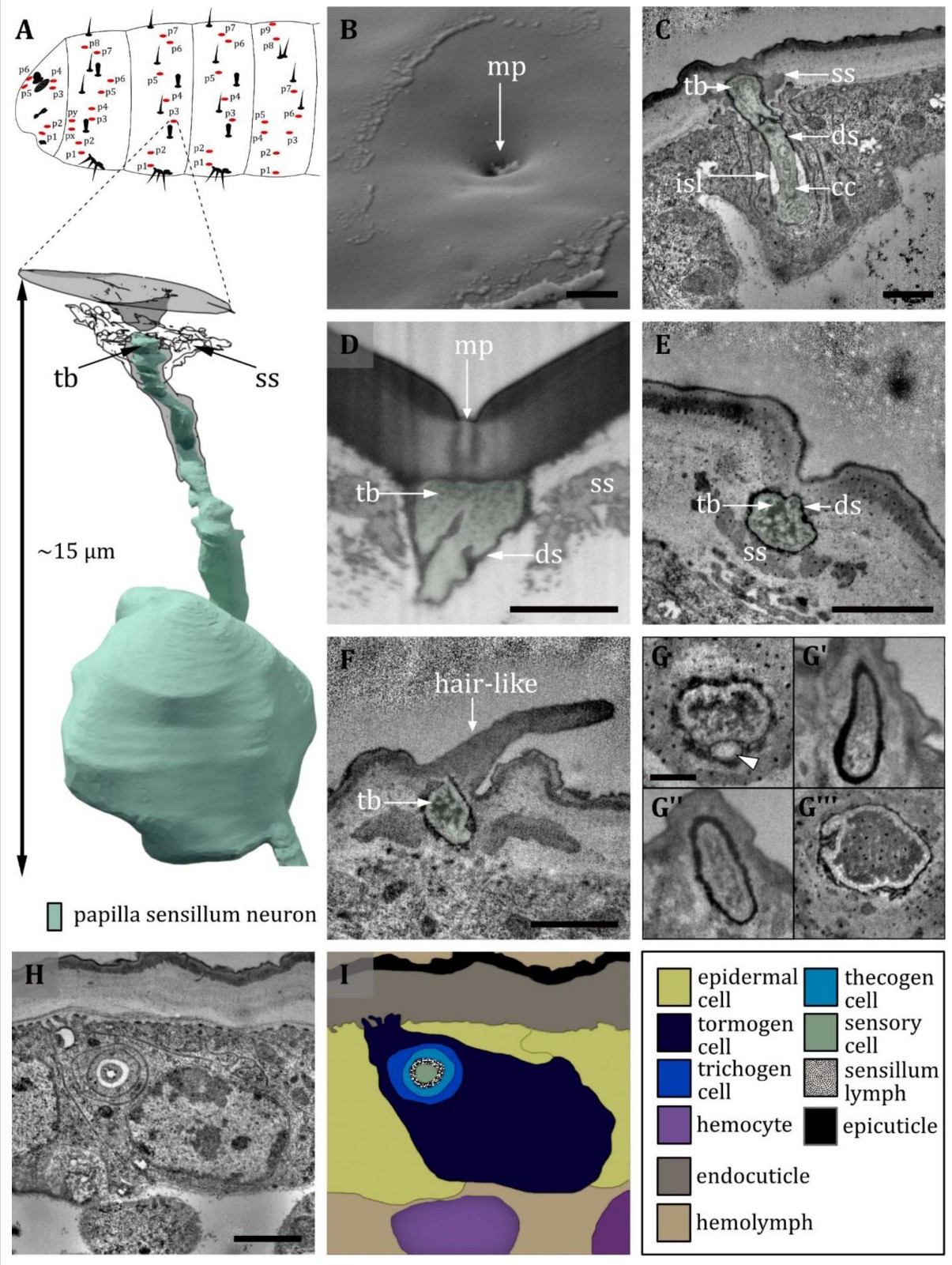

**Figure 10.** Ultrastructure of the papilla sensillum. (**A**) 3D reconstruction of a papilla sensillum of a first third instar (L1) larva (bottom). The outline shows the distribution of this sensillum type on the thoracic and abdominal hemisegments (top). (**B, D, F**) (FIB)-SEM images of third third instar (L3) larvae; (**C, E, G, H**) section scanning transmission electron microscopy (ssTEM) images of L1 larvae. The color code in A applies to all micrographs in this figure. (**B**) Electron micrographs of a papilla sensillum in an L3 larva. The sensillum lays in a cuticular depression with a visible molting pore (mp). (**C**) Longitudinal

*Figure 10 continued on next page*

*Figure 10 continued*

section through a papilla sensillum showing typical features of a mechanoreceptive sensillum. The base is formed by the dendrite, and the tubular body (tb) is enclosed by a dendritic sheath (ds), which is formed by the thecogen cell. The dendrite tip is anchored in the cuticle by the socket septum (ss). Further proximal, the dendrite enters the inner sensillum lymph (isl) cavity and transitions from the outer to the inner dendritic segment at the ciliary constriction (cc). (**D, E**) Longitudinal section through the base of a papilla sensillum in third (**D**) and L1 larva (**E**). A molting pore is visible in (**D**), whereas it is missing in (**E**). (**F**) Untypical papilla sensillum, with a short hair-like protuberance. (**G**) Sensory tip of abdominal papilla sensillum $p_6$ with two dendrites, one without a tubular body (white arrowhead). (**G'**) Sensory tip of abdominal papilla sensillum $p_5$, also called slit papilla sensillum. The tubular body is oval-shaped and appears more electron-lucent than canonical tubular bodies. (**G''**) Sensory tip of thoracic papilla sensillum $p_x$ with an electron-lucent and oval-shaped tubular body. (**G'''**) Sensory tip of thoracic papilla sensillum $p_y$ with very electron-dense tubular body. (**H**) Electron micrograph of sensory and support cells at the level of ciliary constriction. (**I**) Schematic drawing of (**H**) highlighting the sensillar support cells: the thecogen, the trichogen and the tormogen cell. Scale bars: (**B**) 1 µm; (**C**) 0.5 µm; (**D**) 1 µm; (**E**) 1 µm; (**F**) 0.5 µm; (**G**) 0.5 µm; (**H**) 1 µm. Abbreviations: cc - ciliary constriction; ds - dendritic sheath; isl – inner sensillum lymph; mp – molting pore; ss - socket septum; tb - tubular body.

We name these sensilla $VO_1$- $VO_4$. Starting from the medial most sensillum $VO_1$, we number the four sensilla in a clockwise direction in the left and an anti-clockwise direction in the right hemisphere when seen from the front. $VO_1$ forms a shallow dome centering a tiny pore (*Figure 8B*). $VO_1$ is innervated by one dendrite that terminates with a tubular body at the base of the pore and is encased by a dendritic sheath (*Figure 8C, F and G*). The microtubules can be clearly distinguished and are evenly distributed in the tubular body area (*Figure 8C''*). The $VO_1$ sensillum was also termed plate sensillum in earlier literature (*Honda and Ishikawa, 1987*). $VO_2$, $VO_3$, and $VO_4$ lie in pits (*Figure 8*). In L3 larvae, we could observe terminal pores, which are absent in L1 larvae and thus are likely molting pores. $VO_3$ and $VO_4$, like $VO_1$, resemble papilla sensilla. All are innervated by only one dendrite, which composes a tubular body at its tip at the base of the pore openings or the epicuticle, respectively (8 A, C, F, G). The tubular body of $VO_3$ consists of more densely packed microtubules that are not distinguishable from each other (*Figure 8C'*). $VO_4$ displays a unique type of tubular body, which can only be found in this sensillum. It is rather large compared to other tubular bodies, and the dendrite intermingles with electron-dense material of unknown origin in this area (*Figure 8C and D*) (for potential functional implications, please refer to the related section in the discussion). Unlike the other three sensilla, $VO_2$ is innervated by two dendrites (*Figure 8H, I*). Distinguishing it from the dendrites innervating the other sensilla of the VO, the dendrites of $VO_2$ are distally surrounded by a cuticle tube (*Figure 8A and C*), lack a tubular body and branch multifold distally of the ciliary constriction (*Figure 8E*). The terminal pore is also present in L1 larvae. Each of the VO sensilla possesses its individual set of three ACs (*Figure 8F–I*), but the thecogen cell and the trichogen cell of $VO_2$ appear substantially more electron-lucent (*Figure 8H*, asterisks), with mitochondria of altered structure (*Figure 8H*, white arrowhead) compared with ordinary ones (*Figure 8H*, black arrowhead).

## Labial organ

Located on the ventral side of the ventral lip (labium) lays the labial organ (LO – also called lbo) (*Figure 9A and B*). In the present study, we find two sensilla associated with the LO in accordance with *Kankel, 1980*; *Figure 9C, I*. In contrast, *Singh and Singh, 1984* describe three sensilla based on examination of internal ultrastructure. We name the two identified sensilla $LO_1$ and $LO_2$ (*Figure 9C*). $LO_1$ forms a cavity with a pore in the center and sits on a small, shallow socket (*Figure 9C*). The pore is absent in L1 larvae and thus likely a molting pore. $LO_1$ is innervated by one neuron that composes a tubular body on the tip of its dendrite, which terminates below the pore opening. The dendrite is surrounded by a dendritic sheath (*Figure 9E and F*). $LO_2$ forms a knob-like cuticle shaft protruding from the cuticle. The knob's cuticle has a rough texture and appears more electron-lucent than the cuticle of the surrounding body wall (*Figure 9C*). Internally, the knob is filled with an electron-dense material (*Figure 9E*). Like $LO_1$, $LO_2$ is innervated by one dendrite composing a tubular body, which terminates at the base of the knob where a molting pore is present in L3 larvae and absent in L1 larvae (*Figure 9D–F*). The two labial organ sensilla both have their own set of ACs: a thecogen cell, which forms the dendritic sheath; a trichogen cell, which forms the shape of the sensillum; and a tormogen cell, which forms the sensillar socket. We find two more cells in the LO, most likely non-apoptotic glial cells originating from the sensory organ precursor cell (SOP) or its secondary precursor cell (pIIb), respectively (*Fichelson and Gho, 2003*).

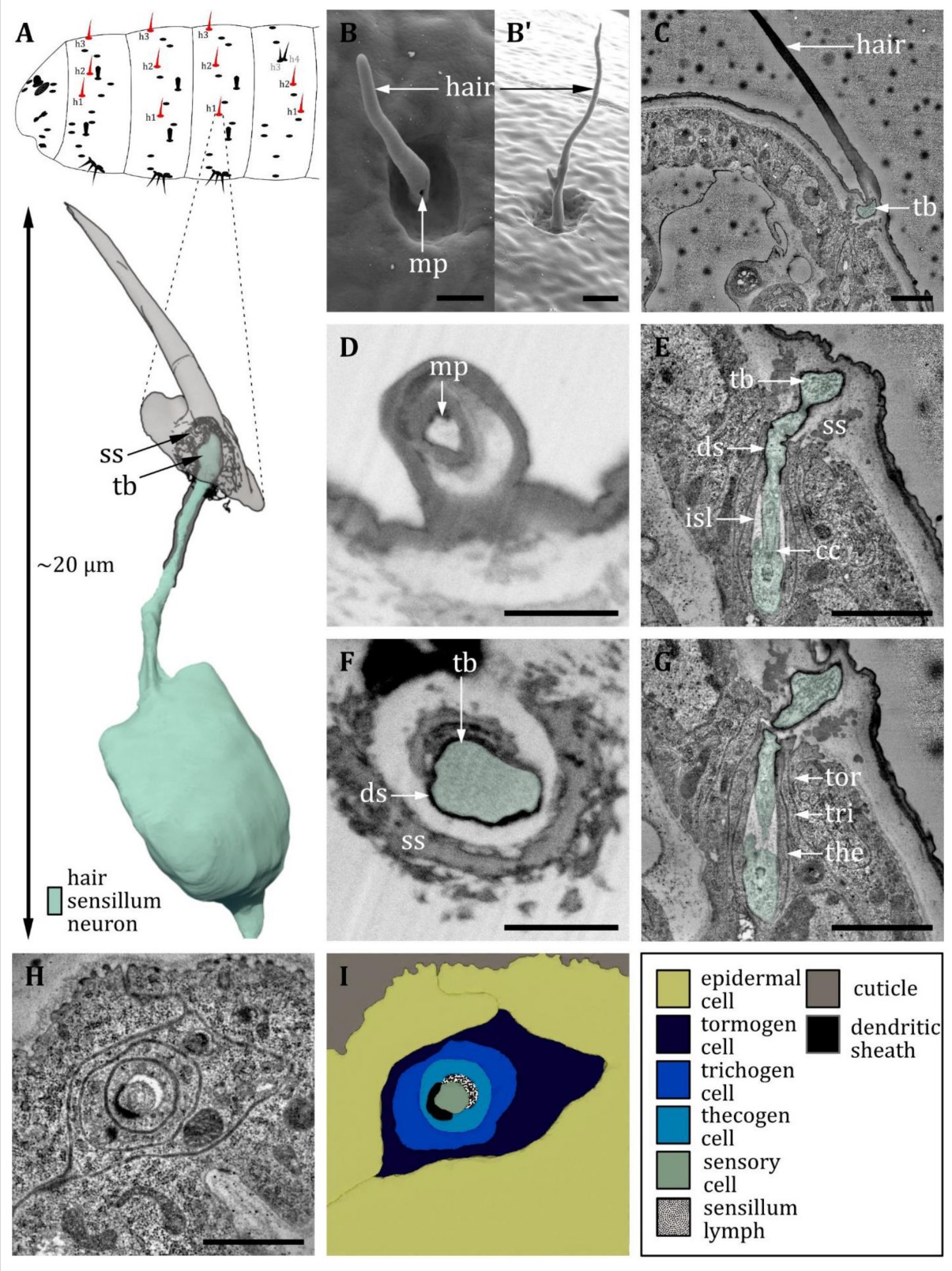

**Figure 11.** Ultrastructure of the hair sensillum. (**A**) 3D reconstruction of a hair sensillum of a first instar (L1) larva (bottom). The outline shows the distribution of this sensillum type on the thoracic and abdominal hemisegments (top). (**B, D, F**) (FIB)-SEM images of third instar (L3) larvae; (**C, E, G, H**) section scanning transmission electron microscopy (ssTEM) images of L1 larvae. The color code in A applies to all micrographs in this figure. (**B, B'**) Electron micrographs of hair sensilla in L3 larvae. Hair sensilla can be branched (**B'**) or unbranched (**B**). A molting pore is visible on the hair in B. (**C**)

*Figure 11 continued on next page*

*Figure 11 continued*

Longitudinal section through a hair sensillum showing typical features of a mechanoreceptive sensillum (see (**E**) for details). The base is formed by the dendrite (green), and the tubular body (tb) is enclosed by a dendritic sheath (ds), which is formed by the thecogen cell. The dendrite tip is anchored in the cuticle by the socket septum (ss). The hair is devoid of dendrites. (**D**) Longitudinal section through the base of a hair with a putative molting pore (mp). (**E, G**) Longitudinal section through the sensory dendrite. The dendritic outer segment enters the lymph cavity and tapers off, reaching the ciliary constriction from where the inner dendritic segment begins. The dendrite is enclosed by the thecogen cell (the), trichogen cell (tri), and tormogen cell (tor). (**F**) Mechanoreceptive region of the dendrite with a tubular body surrounded by a dendritic sheath and enclosed by a septum socket. (**H**) Electron micrograph of sensory and support cells at the level of ciliary constriction. (**I**) Schematic drawing of (**H**) highlighting the sensillar support cells: the thecogen, the trichogen, and the tormogen cell. Scale bars: (**B**) 2 and (**B′**) 1 µm, respectively; (**C**) 1 µm; (**D**) 1 µm; (**E**) 1 µm; (**F**) 1 µm; (**G**) 1 µm; (**H**) 1 µm. Abbreviations: cc - ciliary constriction; ds - dendritic sheath; isl – inner sensillum lymph; mp – molting pore; ss - socket septum; tb - tubular body; the – thecogen cell; to – tormogen cell; tri – trichogen cell.

The online version of this article includes the following figure supplement(s) for figure 11:

**Figure supplement 1.** Mechanotransduction in a hair sensillum.

## Sensilla of thoracic and abdominal segments

### Papilla sensilla

The papilla sensillum is most similar to the canonic type of the campaniform sensillum. Papilla sensilla forms a shallow depression in the cuticlewith a pore in its center in L3 larvae (***Figure 10A, B***). In L1 larvae, the pore is absent and, therefore, a molting pore (***Figure 10C, E***). Papilla sensilla are innervated by one dendrite which terminates with a tubular body below the pore or the epicuticle, respectively (***Figure 10C–E***). The tubular bodies show no organized distribution of microtubules which are difficult to distinguish as they occur in densely packed clusters (***Figure 10E***). The tubular body of the p6 sensillum in the abdominal segments is of a similar shape, although another sensory neuron without a tubular body exists in a shared sensillum space (***Figure 10G***). The dendrites of abdominal sensilla $p_5$, the so-called slit papilla, and the $p_y$ papilla of the first thoracic segment don't show the typical dendritic swelling at the tip and the tubular body appears to be more delicate (***Figure 10G′, G″***). In contrast, the tip of the thoracic $p_x$ neuron is thickened and the tubular body is quite noticeable, as the whole inner area is packed with electron-dense material (***Figure 10G‴***). For all types, the dendrite tip is anchored in the endocuticle by a socket septum (***Figure 10C–E***). The dendrites are enclosed by a dendritic sheath, which is most likely segregated by the thecogen cell (***Figure 10C–E***). The thecogen cell also forms an inner sensillum lymph cavity at the transition from the outer to the inner dendritic segment at the level of the ciliary constriction (***Figure 10C, H, I***). Furthermore, the sensillum is enveloped by the thecogen cell and the tormogen cell (***Figure 10H, I***) . In some exceptional cases, we find a short hair-like structure protruding from the papilla sensilla or positions where we would expect papilla sensilla (***Figure 10F***).

### Hair sensilla

A hair sensillum (***Figure 11A***) is most similar to the canonic type of the trichoid sensillum. Hair sensilla comprises a round, hair-shaped shaft that sits in the center of a shallow cuticle depression (***Figure 11B***). In accordance with previous literature (***Kankel, 1980***), we find that the shaft of hair sensilla varies greatly in size. It might be very short, reduced to a stump, or very long up to more than 15 µm (***Figure 11B***). The form of the hair shaft might vary, too. We observed bifurcated shafts forming two branches (***Figure 11B′***). Furthermore, we observed hair sensilla that come in a pair ('double hair,' ***Figure 12***). The double hair $h_3/h_4$ was exclusively found on abdominal segments. Because of the differences in external morphology of the hair shaft, ***Kankel, 1980*** classified hair sensilla into three different types, called type C, D, and E (***Table 2***). However, we here find that the internal ultrastructure of hair sensilla (***Figure 11C–I***) is in general similar irrespective of the length and the shape of the shaft. The interior of the hair shaft is electron-lucent, surrounded by an electron-dense sheath (***Figure 11C, D***). A pore, presumably a molting pore, can be found at the base of the shaft (***Figure 11D***) in L3 larvae. All hair sensilla are innervated by a single dendrite terminating at the base of the shaft composing a tubular body (***Figure 11E, F***). The tubular body of the hair sensilla is even more thicker than the canonical tubular body of papilla sensilla, and the microtubules are clearly visible and distinguishable from each other. A socket septum is clearly visible (***Figure 11E, F***). Apart from the difference in outer appearance and structure of the tubular body, the hair sensilla are quite similar to the papilla sensilla,

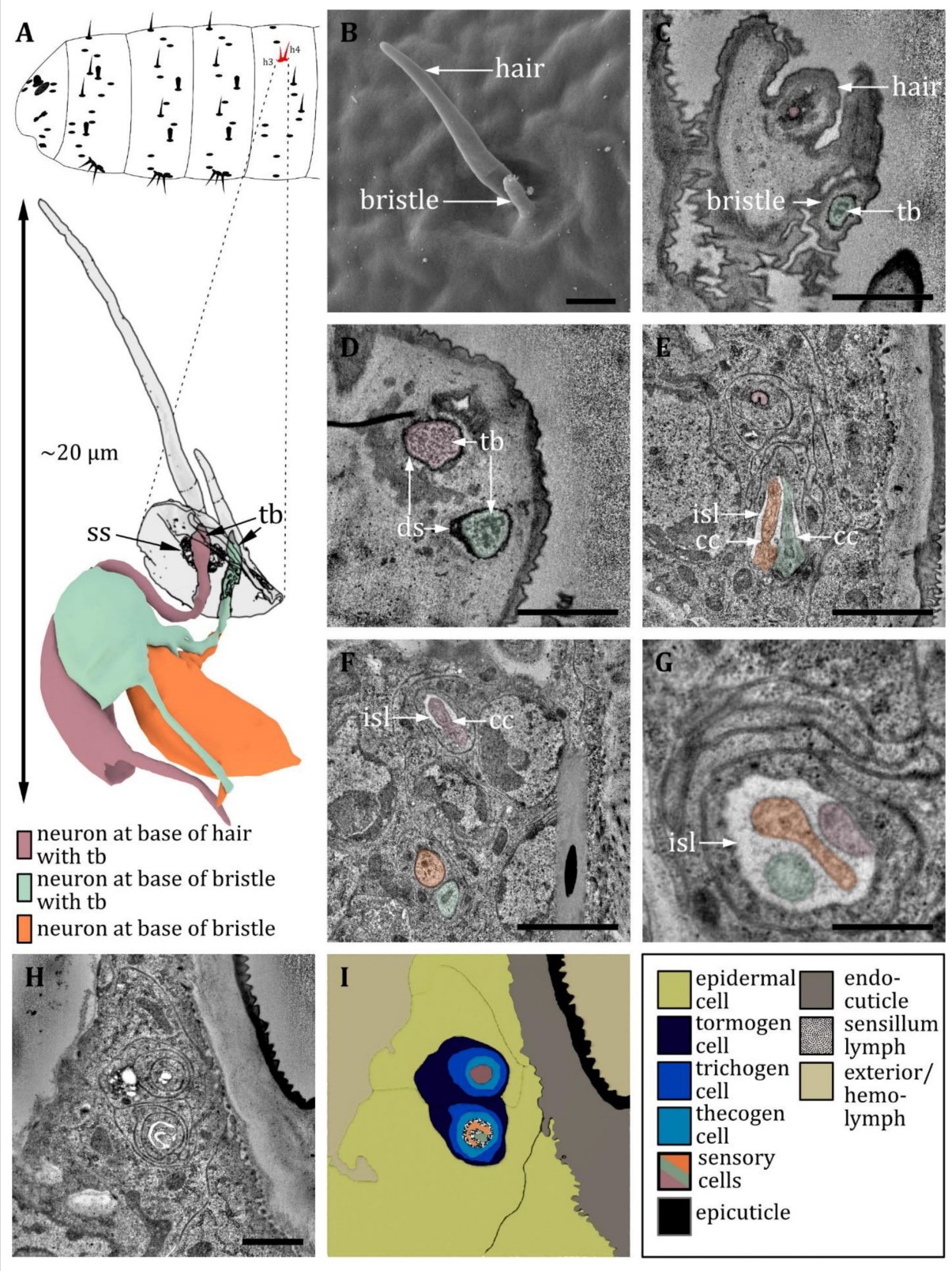

**Figure 12.** Ultrastructure of the double hair organ. (**A**) 3D reconstruction of a double hair organ of a first instar (L1) larva (bottom). The outline shows the distribution of this organ on the abdominal hemisegments (top). (**B**) SEM image of a third instar (L3) larva; (**C-H**) section scanning transmission electron microscopy (ssTEM) images of L1 larvae. The color code in A applies to all micrographs in this figure. (**B**) Scanning electron microscopy (SEM) image of a double hair organ, which consists of a long hair and a short bristle. (**C**) Longitudinal section through a double hair organ showing a dendrite at

*Figure 12 continued on next page*

Figure 12 continued

the base of the hair and a dendrite with a tubular body at the base of the bristle. (**D**) Dendrites at the base of the hair and bristle, both enclosed by a dendritic sheath (ds), and both containing a tubular body but of different appearance. The dendrite at the hair is surrounded by a socket septum. (**E**) ssTEM further proximal of D: the dendrite of the hair is enclosed by a dendritic sheath and support cells. The dendrites of the bristle enter an inner sensillum lymph (isl) cavity and are in the transition from outer to inner segment with associated ciliary constrictions (cc). (**F**) ssTEM further proximal of E: The dendrite of the hair is entering a lymph cavity and is in the transition from outer to inner segment with an associated ciliary constriction. The outer dendritic segments of the bristle-associated sensory cells are visible. (**G**) The dendrites of a double hair organ at the level of the isl cavity: individual variations within one animal can occur; in this case, all three dendrites share the same set of support cells, forming one united sensillum. (**H**) Section through a double hair organ at the level of isl cavity. The enveloping tormogen, trichogen, and thecogen cells can be seen. (**I**) Schematic drawing of H, defining the sensory cells and the associated tormogen, trichogen, and thecogen cells. Scale bars: (**B**) 1 µm; (**C**) 1 µm; (**D**) 1 µm; (**E**) 2 µm; (**F**) 2 µm; (**G**) 0.5 µm; (**H**) 1 µm. Abbreviations: cc - ciliary constriction; ds – dendritic sheath; isl – inner sensillum lymph; tb – tubular body; ss - socket septum.

with a typical set of ACs (*Figure 11G–I*) and a small inner sensillum lymph cavity at the transition from the outer to the inner dendritic segment at the level of the ciliary constriction (*Figure 11E, H, I*).

## Double hair organ

As mentioned previously, the abdominal hair sensilla $h_3$ and $h_4$ represent a special case of compound hair sensilla. It consists of two hairs of different sizes, which are adjacent to each other, sitting in one cuticle depression (*Figure 12A, B*). Usually, the $h_4$ sensillum structure is of the same type as a canonical hair sensillum, containing one sensory cell with a tubular body at the base of the hair (*Figure 12D*). In contrast, the outer hair of $h_3$ is comparatively short and, therefore, called a bristle. The key aspects are similar to the abdominal papilla $p_6$, with two sensory cells (*Figure 12E, F*), one containing a tubular body (which rather exhibits the structural properties of papilla than hair sensilla) (*Figure 12C, D*). Both $h_3$ and $h_4$ possess their own set of ACs and are, therefore, individual sensilla (*Figure 12H, I*), although in one exceptional case, they shared one lymph space (*Figure 12G*). At the level of the ciliary constriction, the thecogen cell forms an inner sensillum lymph cavity (*Figure 12E, F*).

## Knob sensilla

The knob sensillum is most similar to the canonic type of the basiconic sensillum. Knob sensilla are present on thoracic segments (*Figure 13A*) and on the sensory cones of the last abdominal segment (*Figure 16A*), but similar structures can also be found in the terminal organ (*Figure 16G*; *Rist and Thum, 2017*). Knob sensilla have been described under various names such as koelbchen, knob-in-pits, hair-type B, black sensory organs, black dots, sensory papillae, or dorsal/ventral pits (*Hertweck, 1931*; *Lewis, 1978*; *Lohs-Schardin et al., 1979*; *Kankel, 1980*; *Singh and Singh, 1984*; *Campos-Ortega and Hartenstein, 1985*; *Sato and Denell, 1985*; *Dambly-Chaudière and Ghysen, 1986*; *Hartenstein, 1988*; *Campos-Ortega and Hartenstein, 1997*; *Table 2*).

The knob sensilla of the thoracic segments display a common external morphology, a knob-shaped cuticle shaft sunken into a round cuticle cavity (*Figure 13B*). The cuticle shaft slightly protrudes from the cavity. Investigation of internal ultrastructure by targeted FIB-SEM and ssTEM reveals that three dendrites are associated with one thoracic knob sensillum (*Figure 13C–I*). One dendrite innervates the shaft, proceeding to its tip (*Figure 13C*). The other two dendrites end at the base of the knob, one with a tubular body (*Figure 13C–E*). All dendrites are surrounded individually by one dendritic sheath (*Figure 13C–E*), which arises from the common thecogen cell. Each sheath is connected to a pore at the base of the knob in L3 larvae (*Figure 13D*). This pore is not present in L1 larvae (*Figure 13C–E*) and is, therefore, most likely a molting pore. The dendritic sheath of the three dendrites disappears at the epidermal layer. Here, dendrites are surrounded by three common sheath cells (*Figure 13H, I*). In T1, we find one dorsal knob sensillum (dk) and one ventral knob sensillum (vk), whereas in T2-T3, we find one lateral knob sensillum (lk) and one ventral knob sensillum (*Figure 1B* and *Figure 13A*). All thoracic knob sensilla were examined and no difference in structural organization was recognized. In contrast, the knob sensilla of the sensory cones are inconsistent and house either two (*Figure 16D and E*) or three (*Figure 16F*) neurons, whereas the knob sensilla of the TO only contain one sensory cell (*Figure 16G*; *Rist and Thum, 2017*), which dendrite protrudes into the shaft. Knob sensilla of the last abdominal segment are discussed in a separate results section, as they are organized in specialized terminal sense organs.

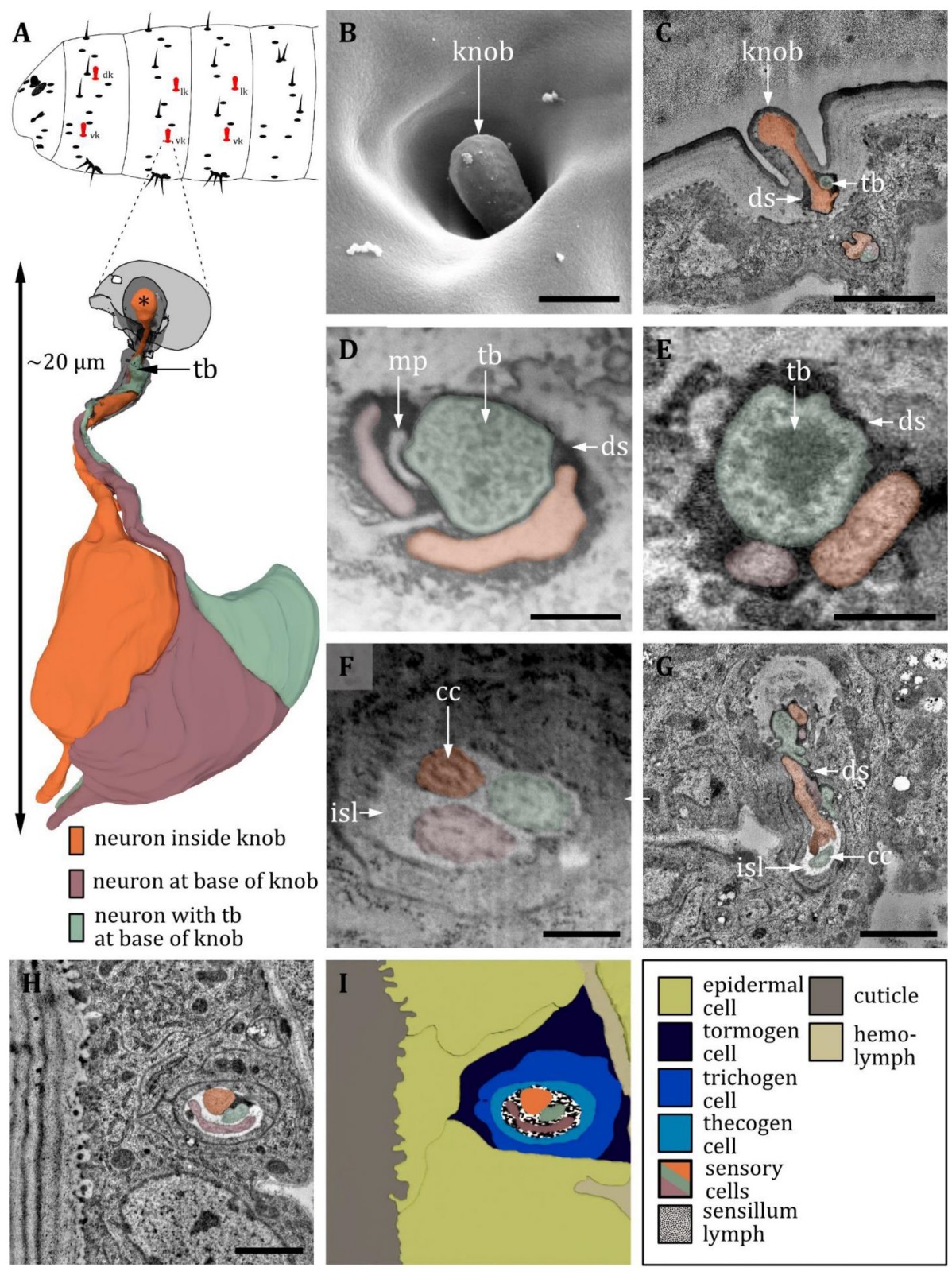

**Figure 13.** Ultrastructure of the knob sensillum. (**A**) 3D reconstruction of a knob sensillum of a first instar (L1) larva (bottom). The outline shows the distribution of this sensillum type on the thoracic hemisegments (top). (**B, D, F**) (FIB)-SEM images of third instar (L3) larvae; (**C, E, G, H**) section scanning transmission electron microscopy (ssTEM) images of L1 larvae. The color code in A applies to all micrographs in this figure. (**B**) Scanning electron microscopy (SEM) image of a knob sensillum. Externally, a knob-shaped sensillum shaft is visible that is sunken into a deep and steep cavity.

*Figure 13 continued on next page*

Figure 13 continued

(**C**) Longitudinal section through the knob sensillum. The dendrite of one sensory cell protrudes into the knob (orange) and bulges out at the end. Two other dendrites, one with (green) and one without (red) a tubular body (tb), end at the base of the shaft. (**D, E**) Close-up view of a cross-section through all three dendrites at the base of the shaft in third (**D**) and L1 (**E**) larvae, respectively. The dendrites are enclosed by a common dendritic sheath (ds). In L3, we see a putative molting pore (mp). (**F**) Cross-section through the three dendrites at the level of the ciliary constriction (cc). The common thecogen cell forms an inner sensillum lymph (isl) cavity. (**G**) Longitudinal section showing all three dendrites from the base of the shaft to their transition from dendritic outer to dendritic inner segment at the cc inside the isl. (**H**) Cross-section with all three dendrites bathed in the isl cavity and enclosed by the tormogen, trichogen, and thecogen cell. (**I**) Schematic drawing of (**H**), defining the sensory cells and the associated tormogen, trichogen, and thecogen cells. Scale bars: (**B**) 0.5 µm; (**C**) 1 µm; (**D**) 1 µm; (**E**) 1 µm; (**F**) 1 µm; (**G**) 1 µm; (**H**) 1 µm. Abbreviations: cc - ciliary constriction; ds - dendritic sheath; isl – inner sensillum lymph; mp – molting pore; tb – tubular body.

## Keilin's organ

The KO is exclusively located on the ventral side of the thoracic segments T1-T3 (*Figure 14A*). The KO is relatively easy to identify and was described for larvae of several dipteran species (*Keilin, 1915*; *Lakes and Pollack, 1990*; *Lakes-Harlan et al., 1991*). The KO of *Drosophila* larvae has so far been described as consisting of three hairs (*Kankel, 1980*). In L1 larvae, we find that two papilla-like sensilla are associated with the organ in addition to three hair sensilla (*Figure 14A, E and G*). One of the papilla-like sensilla is degenerating during the L1 stage (*Figure 14E–F*). Signs of degeneration are the comparatively small diameter of the dendrite (*Figure 14E*) and poor axonal development, like the absence of growth cones and axon branching (not shown). This finding is congruent with a study of the KO of the larvae of *Phormia* (*Lakes-Harlan et al., 1991*) and provides an explanation for studies that find five sensory neurons innervating the KO of *Drosophila* larvae (*Dambly-Chaudière and Ghysen, 1986*; *Campos-Ortega and Hartenstein, 1997*). In L3 larvae, the (surviving) papilla sensillum is externally recognizable by a tiny pore in the cuticle (*Figure 14B*). Also, a pore is found on the base of each of the hairs (*Figure 14D*). Because these pores are not found in L1 larvae, they are most likely molting pores. Investigation of the internal ultrastructure reveals that each of the three hair sensilla and the two papilla sensilla are associated with a single dendrite. All dendrites terminate with a tubular body below the cuticle; this organization represents the standard types of hair or papilla sensilla (*Figure 14E and F*). Also, all KO sensilla possess their own set of enveloping ACs (*Figure 14H, I*).

## Sensilla of the anal division
### Terminal sensory cones

The terminal sense organs or sensory cones are located at the fused terminal segments (*Figure 1A*, *Figure 15A*, *Figure 16A*). According to the literature (*Whittle et al., 1986*; *Denell and Frederick, 1983*; *Dambly-Chaudière and Ghysen, 1986*; *Jürgens, 1987*), we could identify seven distinct sensory cones $t_1 – t_7$. They either house a knob sensillum ($t_3$, $t_5$, $t_7$), a hair sensillum ($t_4$, $t_6$), both ($t_2$), or both plus an additional papilla sensillum at the base of the cone ($t_1$) (*Figure 16—figure supplement 1*). These results are consistent with previous findings in L1 larvae (*Dambly-Chaudière and Ghysen, 1986*; *Jürgens, 1987*), although the number of corresponding neurons is slightly different. We found that knob sensilla of $t_1$, $t_5$, and $t_7$ are only innervated by two sensory neurons (*Figure 16E*) in contrast to three sensory neurons that were reported in the embryo (*Dambly-Chaudière and Ghysen, 1986*) and that we find in $t_2$ and $t_3$ (*Figure 16C*). These knob sensilla are still innervated by one neuron protruding into the shaft and one neuron containing a tubular body that sits at the base of the shaft. A third neuron without a tubular body is absent (*Figure 16E*). The hair sensilla associated with the sensory cones are innervated by one sensory cell with a tubular body on its tip (*Figure 15B*) and exhibits similar features as the canonic hair sensilla of the body wall. The only papilla sensillum associated with the sensory cones sits at the base of the $t_1$ and shares the same structure as the canonic papilla sensilla. The $t_1$ lays in close proximity to the anal plate towards the caudal end. $T_2$ is situated dorso-caudal of $t_1$, halfway in between the anal plate and the posterior spiracle. $t_3$ lays anterodorsally of $t_2$ and the cone of $t_4$ lays anterodorsally of $t_3$. The cone of $t_5$ is located dorsal of $t_4$, whereas $t_6$ is located closer to the dorsal midline than $t_5$. The cone of $t_7$ is located at the dorsal midline and at the base of the posterior spiracle. All knob sensilla of the terminal sensory cones show a similar external structure as thoracic knob sensilla, but their shafts protrude far out of the cavity in L1 larvae (*Figure 16B and C*; *Figure 16—figure supplement 1*). In L3 larvae, the sensory cones further change in appearance, and the knob and hair sensilla are sunken into the full-grown cone and surrounded by

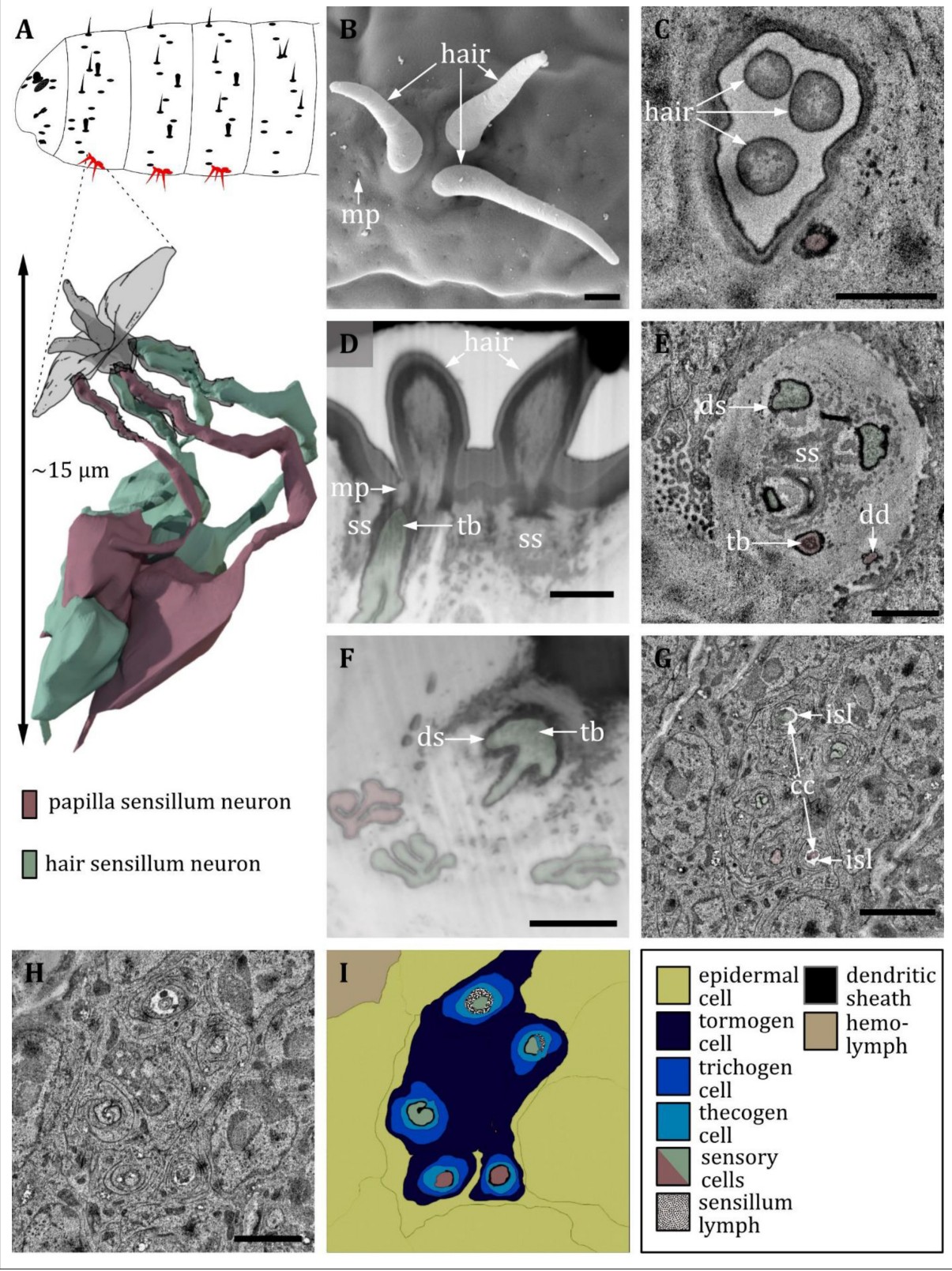

**Figure 14.** Ultrastructure of the keilin's organ. (**A**) 3D reconstruction of a keilin's organ of a first instar (L1) larva (bottom). The outline shows the distribution of this sensillum type on the thoracic hemisegments (top). (**B, D, F**) (FIB)-SEM images of third instar (L3) larvae; (**C, E, G, H**) section scanning transmission electron microscopy (ssTEM) images of L1 larvae. The color code in A applies to all micrographs in this figure. (**B**) Scanning electron microscopy (SEM) image of a keilin's organ consisting of three external hairs. (**C**) Cross-section of keilin's organ with three hairs of hair-like sensilla and

*Figure 14 continued on next page*

*Figure 14 continued*

one dendrite of a papilla-like sensillum without a hair. (**D**) A dendrite with a tubular body (tb) and a putative molting pore (mp) is visible at the base of the left hair. The dendrite is surrounded by a dendritic sheath and enclosed by a septum socket (ss). (**E**) The keilin's organ dendrites have visible tubular bodies but absent molting pores. The dendrites are surrounded by a dendritic sheath and enclosed by a septum socket. Three hair-like and two papilla-like sensilla can be observed. (**F**) Keilin's organ dendrites with visible tubular bodies. In L3 larvae, only one papilla-like sensillum is abundant. (**G**) The dendrite of the keilin's organ is further proximal than (**E**), with ciliary constrictions (cc) at the transition from dendritic outer to dendritic inner segment. Dendrites are surrounded by thecogen cells, which form an inner sensillum lymph (isl) cavity in this region. (**H**) Cross-section further proximal than (**G**). The enveloping tormogen, trichogen, and thecogen cells can be seen. (**I**) Schematic drawing of (**H**), defining the sensory cells and the associated tormogen, trichogen, and thecogen cells. Scale bars: (**B**) 1 µm; (**C**) 1 µm; (**D**) 1 µm; (**E**) 1 µm; (**F**) 1 µm; (**G**) 2 µm; (**H**) 2 µm. Abbreviations: cc - ciliary constriction; dd – dendritic degeneration; ds - dendritic sheath; isl – inner sensillum lymph; mp – molting pore; ss - socket septum; tb – tubular body.

broad-based apposing leaflets (*Kuhn et al., 1992*). All sensilla associated with the terminal sensory cones display a typical sensilla configuration: their dendrites are bathed in the inner sensillum lymph and ensheathed by a typical set of support cells (*Figure 15C and D* and *Figure 16F, H, I*). Besides the prominent sensory cones, there are some less noticeable external sensilla.

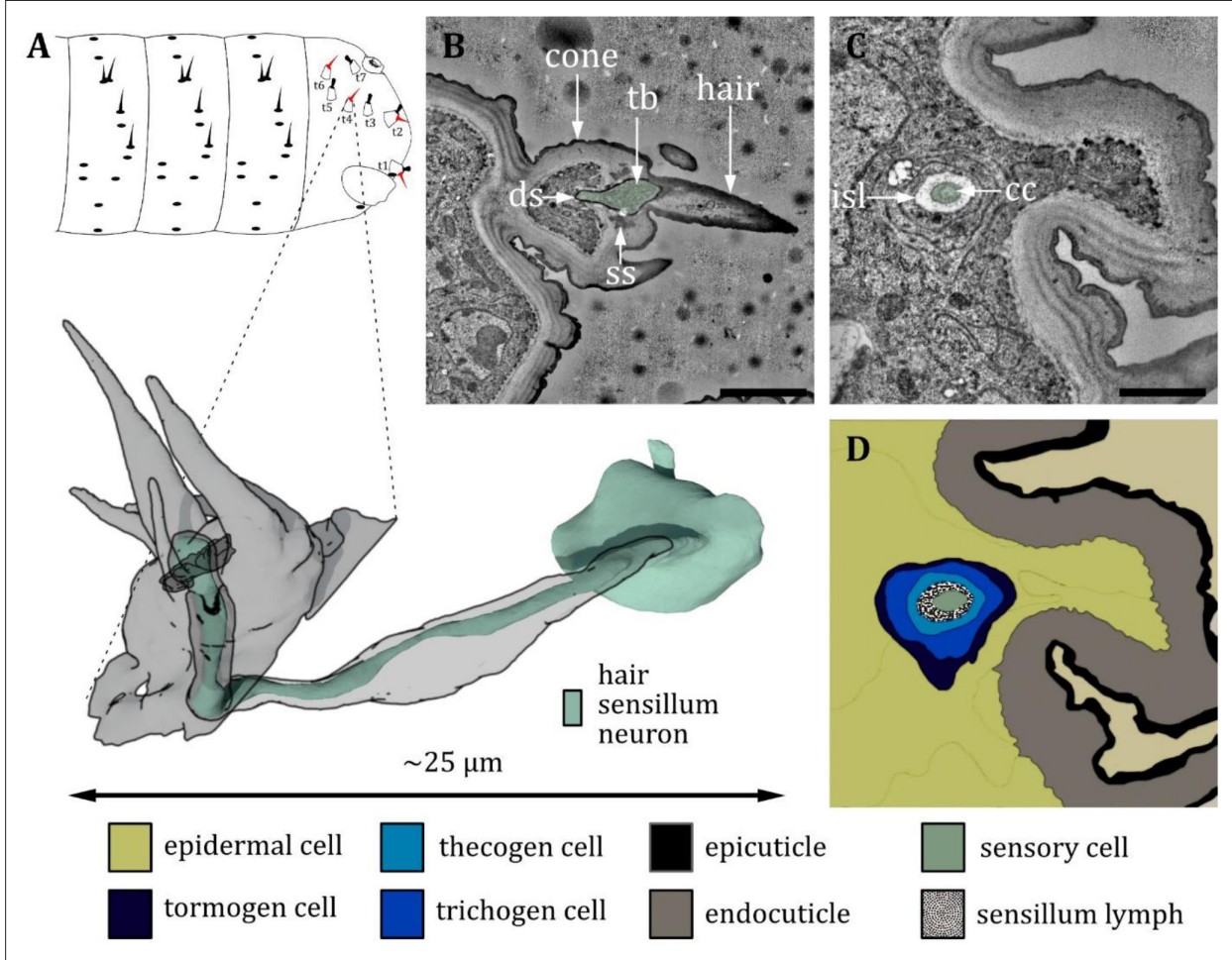

**Figure 15.** Ultrastructure of the hair sensillum at the terminal sensory cones. (**A**) 3D reconstruction of a hair sensillum at the terminal segment of a first instar (L1) larva (bottom). The outline shows the distribution of this sensillum type on the last fused hemisegments (top). (**B, C**) section scanning transmission electron microscopy (ssTEM) images of a third instar (L1) larva. The color code in A applies to all micrographs in this figure. (**B**) Longitudinal section through the hair sensillum. A sensory hair is sitting on top of a cone-like structure with more (non-innervated) hairs and bristles. A dendrite with a tubular body (tb) ends at the base of the sensory hair. It is surrounded by a dendritic sheath (ds) and anchored in the cuticle by a septum socket (ss). (**C**) Longitudinal section further proximal than B at the level of the ciliary constriction (cc). The inner sensillum lymph (isl) cavity and the support cells can be observed. (**D**) Schematic drawing of (**C**), defining the sensory cell and the associated tormogen, trichogen, and thecogen cells. Scale bars: (**B**) 2 µm; (**C**) 1 µm. Abbreviations: cc - ciliary constriction; ds - dendritic sheath; isl – inner sensillum lymph; ss - socket septum; tb – tubular body.

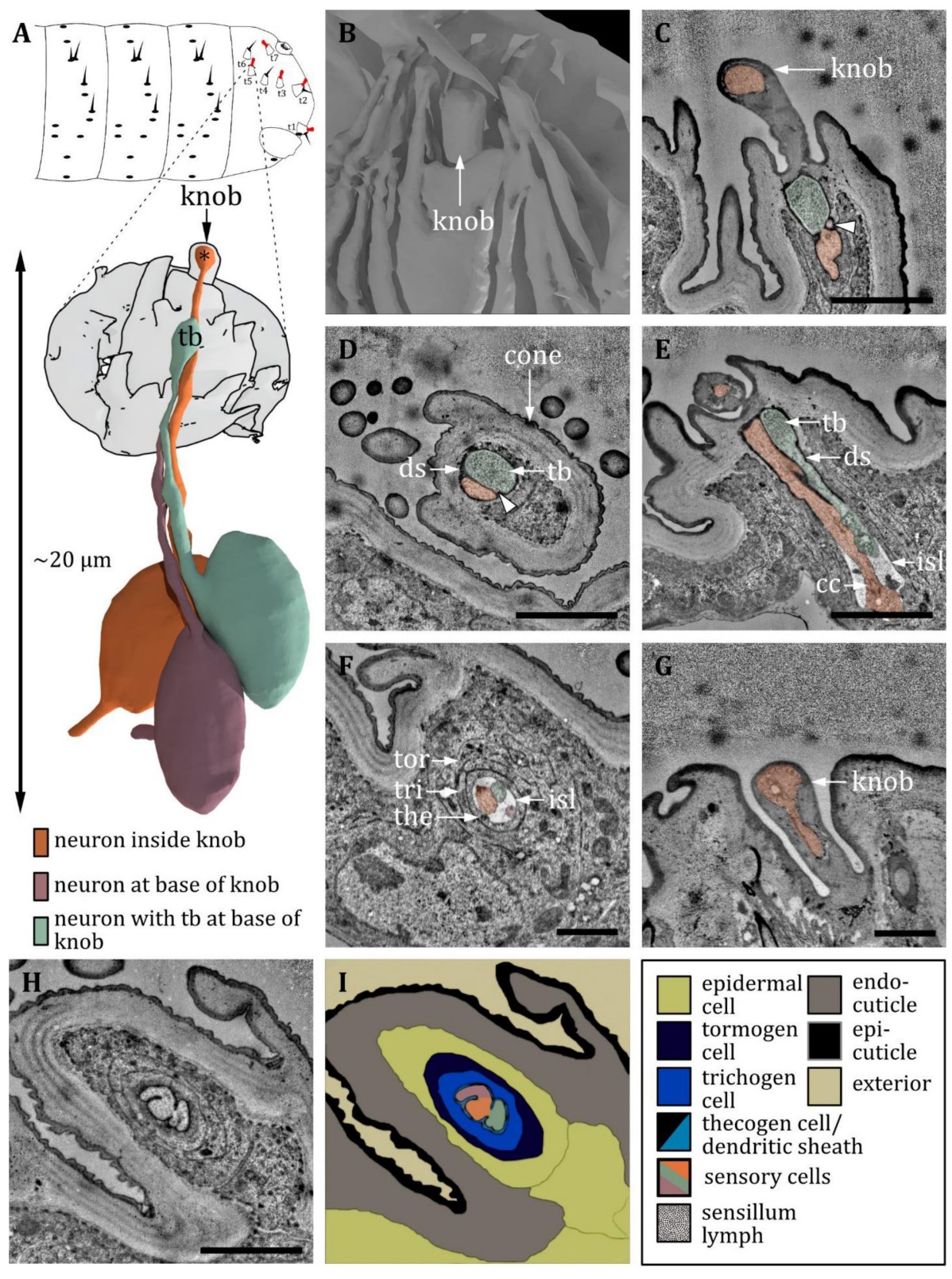

**Figure 16.** Ultrastructure of the knob sensillum at the terminal sensory cones. (**A**) 3D reconstruction of a knob sensillum at the terminal sensory cone t₃ in a first instar (L1) larva (bottom). The outline shows the distribution of knob sensilla on the last fused hemisegments (top). (**C-H**) Section scanning transmission electron microscopy (ssTEM) images of an L1 larva. The color code in A applies to all micrographs in this figure. (**B**) 3D reconstruction of the outer appearance of t₃. Inside the cone, a knob-shaped sensillum shaft is visible that is sunken into a deep and steep cavity. The cone is surrounded

*Figure 16 continued on next page*

*Figure 16 continued*

by (non-sensory) hairs and bristles. (**C, D**) Longitudinal section through the knob in the cone $t_3$. The dendrite of one sensory cell protrudes into the shaft and bulges out at the end (orange). Two other dendrites end at the base of the shaft, one containing a tubular body (green), the other not (red, arrowhead). All dendrites are enclosed by a dendritic sheath (ds). Knob sensilla innervated by three sensory cells are found in $t_2$ and $t_3$. (**E**) Longitudinal section through the knob sensillum of cone $t_5$. Only two dendrites are abundant, one protruding into the sensillum shaft (orange) and the other containing a tubular body (green). Further proximal, the dendrites are bathed inside the inner sensillum lymph (isl) at the level of the ciliary constriction (cc). Knob sensilla innervated by only two sensory cells are found in cones $t_1$, $t_5$, and $t_7$. (**F**) Longitudinal section through the three dendrites of $t_3$ at the level of the ciliary constriction. The common thecogen cell (the) forms an isl cavity. The thecogen cell is enclosed by the trichogen cell (tri) and the tormogen (tor) cell. (**G**) Knob sensillum of the terminal organ with only one abundant dendrite (orange) that protrudes into the knob. (**H**) Longitudinal section of $t_3$ proximal of (**D**) with all three dendrites sharing a common dendritic sheath, enclosed by the tormogen, trichogen, and thecogen cell. (**I**) Schematic drawing of (**H**), defining the sensory cells and the associated tormogen, trichogen, and thecogen cells. Scale bars: (**C**) 2 µm; (**D**) 2 µm; (**E**) 2 µm; (**F**) 1 µm; (**G**) 1 µm; (**H**) 1 µm. Abbreviations: cc - ciliary constriction; ds - dendritic sheath; isl – inner sensillum lymph; tb – tubular body; the – thecogen cell; to – tormogen cell; tri – trichogen cell.

The online version of this article includes the following figure supplement(s) for figure 16:

**Figure supplement 1.** 3D reconstruction of terminal sensory cone 1 (t1).

## Spiracle sense organ

The spiracle sense organ (sp) consists of four sensilla ($sp_{A-D}$) which are located in close proximity to the spiracular openings at the base of the spiracular hairs (*Figure 17A–D*). They are all single-innervated, contain a tubular body (*Figure 17D*), and display a canonic set of support cells (*Figure 17E and F*). The spiracular glands lay in close proximity to these sensilla (*Figure 17E and F*).

## Ventral anal papilla sensillum

One papilla sensillum can be found close to the anal plates. It is located at the anal opening and called ventral anal papilla sensillum (vas) in the literature (*Campos-Ortega and Hartenstein, 1997*). In contrast to Hartenstein's findings, we find it to be only single-innervated. Notably, one unpaired anal cone or anal tuft (*Lohs-Schardin et al., 1979*; *Denell and Frederick, 1983*) is located in close proximity and was mistaken as a sensory cone before. We could not find any sensory cell associated with this structure, although vas lies in close proximity (*Figure 1B*).

Overall, we obtained a complete picture of all larval external sensory structures and their associated sensory neurons. In the following, we discuss their putative sensory mechanism and function according to their internal morphology and ultrastructure and classify the different sensilla and their associated neurons.

## Discussion

We present a comprehensive anatomical analysis of external sensory organs of the head, thoracic, and abdominal segments of the *Drosophila* larva. The application of optimized FIB-SEM and ssTEM allowed us to image and reconstruct small sensilla as well as large and complex sensory organs. Our experimental FIB-SEM approach can easily be adapted to study the morphology of external sensilla and other external structures of further insect species. This methodology is a prerequisite to investigate the different characteristics of insect peripheral nervous systems in a comparative way. With respect to *Drosophila* larvae, our work provides fundamental anatomical insights necessary to further decipher the relation between the structure and function of sensory organs. Certainly, a future key step would be to establish transgenic fly lines that label the sensory neurons of the sensilla, allowing the study of their functionality through molecular, behavioral, and physiological experiments. This might be possible through a variety of genetic driver lines from different Gal4, LexA, and Q system collections (*Lewis, 1978*; *Brand and Perrimon, 1993*; *Lai and Lee, 2006*; *Pfeiffer et al., 2010*; *Li et al., 2014*; *Gohl et al., 2011*; *Tirian and Dickson, 2017*). In addition, there are even more specific lines based on multiple overlapping promoter expression patterns such as the split-Gal4 system (*Luan et al., 2006*). Unfortunately, these lines have mostly been studied only with respect to their expression in the central nervous system, and anatomical studies of the peripheral nervous system are limited. It will be equally important to connect the described peripheral sensory system of the larva with the already existing brain connectome in order to better understand the processing of external environmental information and its relevance for behavior

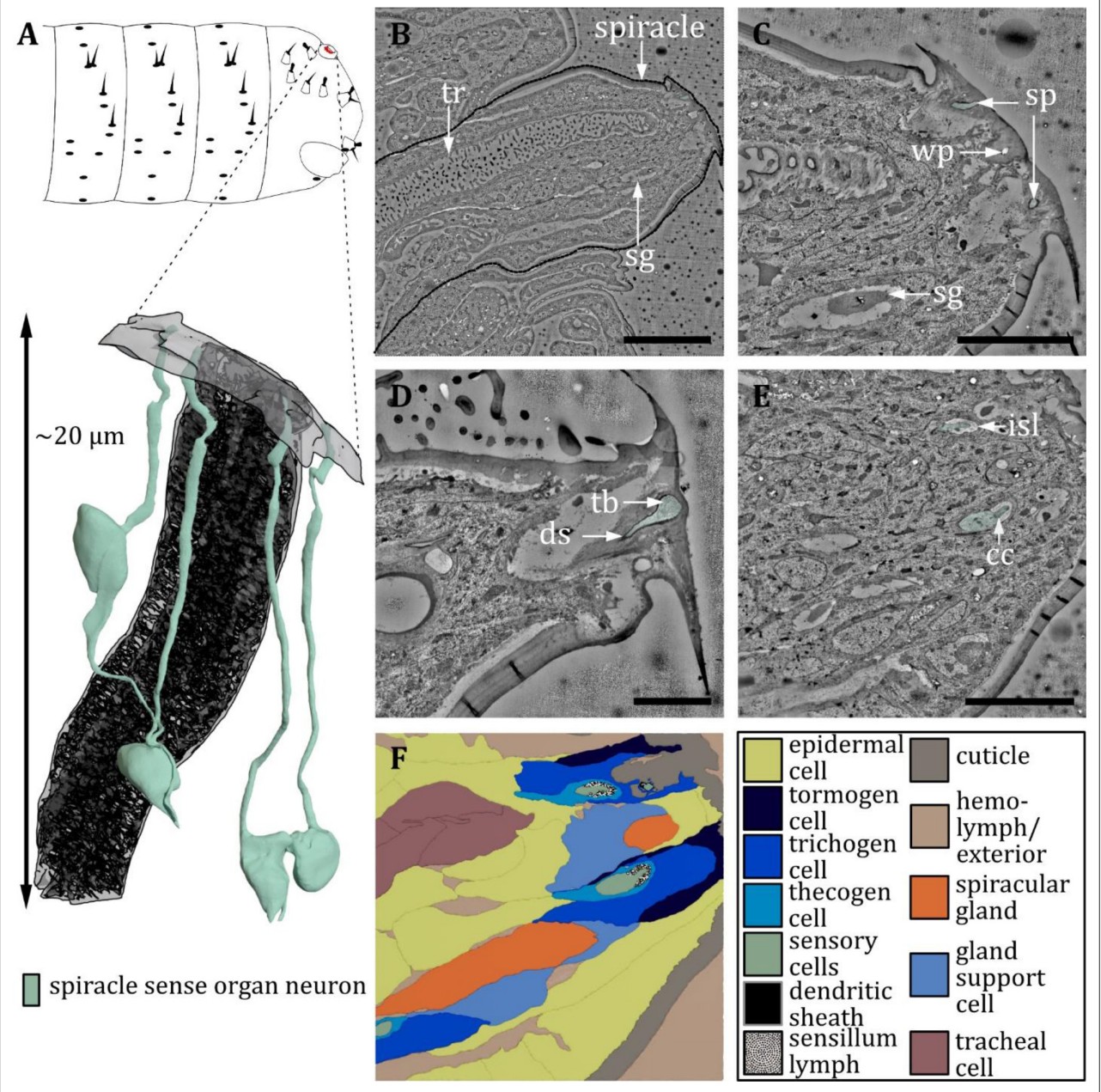

**Figure 17.** Ultrastructure of the spiracle sense organ. (**A**) 3D reconstruction of the spiracle sense organ (sp) at the terminal segment of a first instar (L1) larva (bottom). The outline shows the position of this organ on the last fused hemisegments (top). (**B, C**) Section scanning transmission electron microscopy (ssTEM) images of an L1 larva. The color code in A applies to all micrographs in this figure. (**B**) Longitudinal section through a posterior spiracle. The main tracheal tube (tr) and two spiracular glands (sg) are visible. (**C**) Longitudinal section of the tip of the spiracle. Two sensilla of the spiracle sense organ can be observed. Furthermore, a spiracular gland and a wax pore (wp) associated with another spiracular gland are visible. The glands secrete wax around the tracheal valve to prevent moisture ingress. (**D**) Close-up view of the mechanoreceptive region of a spiracle sensillum near the tracheal valve. The dendrite contains a tubular body (tb) at the tip and is enclosed by a typical dendritic sheath (ds). (**E**) Longitudinal section of the tip of the spiracle further proximal than (**B**) showing the two sensilla inside the inner sensillum lymph (isl) with their associated support cells. The ciliary constriction (cc) is visible for one sensillum. (**F**) Schematic drawing of (**E**), defining the sensory cells and the associated tormogen, trichogen, and thecogen cells. Furthermore, the tracheal and gland cells are shown. Scale bars: (**B**) 10 μm; (**C**) 5 μm; (**D**) 2 μm; (**E**) 5 μm. Abbreviations: cc - ciliary constriction; ds - dendritic sheath; isl – inner sensillum lymph; sg – spiracular glands; sp – spiracle; tb – tubular body; tr – trachea.

(*Hückesfeld et al., 2021*; *Miroschnikow et al., 2018*; *Eichler et al., 2017*; *Eschbach et al., 2020*; *Ohyama et al., 2015*). Such linkage would be important, for example, to analyze whether mechanosensory information from different body segments converges in a particular brain region and forms a stable, spatially conserved map of inputs - similar to the olfactory system (*Ramaekers et al., 2005*; *Berck et al., 2016*). Similarly, it would allow us to understand for the first time how taste information from different external sensory organs is interconnected with the internal pharyngeal and enteric nervous systems.

The standard configuration of the larval body wall covers a total of 541 sensory cells associated with the external sensilla. 363 are probably mechanosensory according to their internal structure, as they possess a tubular body. The 363 mechanosensory cells can further be grouped into 201 neurons belonging to papilla sensilla, 100 to hair sensilla, and 22 to knob sensilla. The remaining 40 mechanosensory cells can be found in the major head organs and in the spiracle sense organ. We find 42 olfactory and 12 thermosensory neurons in the DOs and 50 gustatory neurons of which 46 cells are distributed in the TOs and 4 in the VOs. We find additional 26 sensory neurons in knob sensilla that are probably important for oxygen and carbon dioxide sensing. Their dendrites protrude into the sensillum shaft, where the dendritic sheath is perforated and, therefore, susceptible to chemical compounds that are able to diffuse into the cuticle. We also found 48 sensory cells whose ultrastructure was insufficient to make well-grounded predictions about their sensory mechanism, as they lacked noticeable structural components. These cells are distributed evenly throughout the larval body. We find two in each of the DOs, one in each thoracic knob sensillum, one each in the abdominal papilla sensilla $p_6$, one in the abdominal hair sensilla $h_3$, and one each in the knob sensilla of the terminal sensory cones $t_2$ and $t_3$. Apart from these sensory cells, there are more to be found in the larval body, like in chordotonal organs, pharyngeal organs, or multidendritic neurons. These were not regarded in this study but will be the subject of future studies.

## Organization of larval sensilla types

### Nomenclature, variability, and ultrastructure

As mentioned before, various names were given to the sensilla, not only immature insects but also larval *Drosophila*. Here, we aimed to list all names given to them (*Tables 1 and 2*) and to establish a standard nomenclature for *Drosophila* larval sensilla, using our detailed and comparative approach as a basis for the classification of all external sensilla. We identified three major types of sensilla which are most abundant on the larval body wall: papilla sensilla, hair sensilla, and knob sensilla. They occur either as solitary sensilla or are integrated into sensory organs or compound sensilla like the terminal organ, the keilin's organ, or the terminal sensory cones. Also, they exhibit slight differences in terms of associated sensory cells or external appearance. Besides these most abundant types, we found specialized (olfactory and gustatory) sensilla, mainly in the organs of the head region.

Taking advantage of the whole larval volume, we were able to identify and name all neurons associated with the external sense and chordotonal organs (*Figure 1C*). Again, we used the nomenclature of Dambly-Chaudière and Ghysen (external sense organs) and Campos-Ortega and Hartenstein (chordotonal organs) but made slight modifications to incorporate deviating cell numbers and to create a consistent nomenclature that can be used in databases.

These changes are: (i) for all external sensilla, the associated neurons are named in the following pattern: position within the segment (dorsal (d), lateral (l), ventral 1 (v), ventral 2 (v'), ventral 3 (v'')), type of sensillum (external sensilla/sense organs (es) or chordotonal organs (Ch)), number of associated neurons (1, 2, 3,…) and order of appearance from ventral to dorsal (A, B, C, …); (ii) for the fused head and abdominal segments, we also used this pattern for simplification and clarity, even though we thereby ignore the segmental borders; (iii) to avoid confusion, we did not assign names that were used for other neurons again, for example, the neurons of abdominal hair sensillum $h_3$ were named $des_A$ before, as *Dambly-Chaudière and Ghysen, 1986* only could identify one neuron in their camera lucida drawings in the embryo. On the contrary, we did always see two neurons innervating $h_3$, and, therefore, called these $des_{2A}$. The remaining dorsal external sense neurons did not change in naming, even though $des_{1A}$ is not assigned. A full list of the external sense neurons is given in *Table 2*. In the following, we discuss different criteria used to assign putative functions to the identified sensilla and sensory cells.

## Tubular bodies

Within the larval sensilla, we find a variation in the shape and size of tubular bodies, a well-known structure regarded to mediate mechanical sensation in insect sensilla sensory neurons (*Thurm, 1964*; *Keil, 1997*). These variations likely correlate with different mechanisms of mechanotransduction. For example, the tubular body of the hair sensilla is shaped in a way to transduce directional force applied to the hair (see *Figure 11—figure supplement 1*). When the hair is deflected, pressure is either applied or released from the dendrite tip via the cuticular structures (like a lever). In papilla sensilla, the force is applied unidirectional, either through deformation of the cuticle through external touch or movement of the body wall during motion. Abdominal $p_5$ papilla are contained in vertical slits (slit papilla), and the tubular body sits beneath an elongated, cuticular mound in the center of the slit (*Figure 10G' G"*). Putatively, the tubular body is deformed by compression of the slit during movement. In contrast to the other tubular bodies (and especially to the delicate tubular bodies of $p_5$ and $p_y$ sensilla), the tubular body of $p_x$ at the first thoracic segment appears to be very dense (*Figure 10G"'*). It can be inferred that a compact tubular structure is also more rigid, leading to an increased sensitivity of the mechanotransduction apparatus, as it acts as an abutment for the spring-like proteins that work as mechanotransducers. However, additional physiological data on these sensory neurons is required to substantiate this hypothesis linking structure and function.

## Developmental aspects

Morphological studies on the ultrastructure of the sensory neurons in *Drosophila* were either executed on *Drosophila* embryos (*Campos-Ortega and Hartenstein, 1997*; *Dambly-Chaudière and Ghysen, 1986*), first (*Hartenstein, 1988*), second (*Singh and Singh, 1984*) or third instar larvae (*Dambly-Chaudière and Ghysen, 1986*). Over the years, these works led to contradictory assumptions about the number and location of the larval sense organs and their corresponding sensory cells. In addition, various functions were to some extent assigned to the sensilla. Our analysis, using L1 and L3 larvae, allowed for clarification of several ambiguities. Primarily, we could address the difference between functional pores and molting pores. Pores that were found in L3 larvae but absent in L1 larvae were considered to be traces of molting and, therefore, not necessary for the sensory function of these sensilla. In most cases, this was true for mechanosensitive sensilla that are widely distributed on the larval body wall. Given their pore-like appearance in L3, especially papilla sensilla were suspected to serve chemosensory function in the past. Additionally, the sensilla of the KO were assumed to serve different sensory functions before (*Hafez, 1950*; *Benz, 1956*) and some of the peripheral sensilla of the DO were termed lateral pore receptors (*Singh and Singh, 1984*; *Chu and Axtell, 1971*) or classified as contact chemoreceptors (*Chu and Axtell, 1971*). If pores are absent in the sensilla of L1 larvae, they probably do not serve chemosensory function, although we cannot rule out this function completely given the lack of physiological experiments. On the other hand, pores that were found in both instars were considered to be functional pores and, therefore, most likely associated with external gustatory sensilla. Pore-containing sensilla are limited and restricted to the $VO_2$ of the ventral organ (*Figure 8C and E*) and the pit and papilla sensilla of the terminal organ (data not shown; see results of *Rist and Thum, 2017*).

Second, we could address the open questions regarding the number of sensilla and sensory neurons. For example, *Singh and Singh, 1984* reported five sensilla in the VO, but our results show only four sensilla in L1 as well as in L3 larvae. Furthermore, they reported five dendrites in $VO_2$. Because we obtained volumes of the VO including the sensory cell bodies, we could show that $VO_2$ is innervated by two sensory cells with one proximal dendrite each, which branch multifold when entering the pore channel. In the abdominal double hair, the $h_3$ sensillum houses two sensory cells instead of one reported before (*Campos-Ortega and Hartenstein, 1997*; *Dambly-Chaudière and Ghysen, 1986*). The keilin's organ was reported to house either four (*Lakes-Harlan et al., 1991*) or five (*Campos-Ortega and Hartenstein, 1985*; *Campos-Ortega and Hartenstein, 1997*; *Dambly-Chaudière and Ghysen, 1986*; *Lakes-Harlan et al., 1991*) sensilla previously. With our work, we could explain these differences: one papilla sensillum is still abundant in L1 larvae, but already shows signs of degeneration and is completely missing in L3 larvae (*Figure 14F*). This is the only example of age-specific degeneration of sensilla between larval moltings that we could identify.

# Hypothetical functions of larval head, thoracic, and abdominal sensilla

## Dorsal organ

In *Drosophila*, the six peripheral sensilla of the DO were hypothesized to serve different modalities: contact chemosensation (gustation), mechanosensation, thermosensation, and hygrosensation (*Chu and Axtell, 1971*; *Python and Stocker, 2002*; *Kwon et al., 2011*; *Klein et al., 2015*; *Ni et al., 2016*; *Hernandez-Nunez et al., 2021*). Evidence derives from observed ultrastructure indicating certain functions (*Chu and Axtell, 1971*), and more recently from anatomical, physiological, and behavioral experiments (*Kwon et al., 2011*; *Klein et al., 2015*; *Ni et al., 2016*; *Hernandez-Nunez et al., 2021*). We conclude that $DO_{p3}$, $DO_{p5}$, and $DO_{p6}$ are candidates for thermosensory function, which was confirmed recently (*Hernandez-Nunez et al., 2021*). Typically, insect thermosensory sensilla are associated with neurons that terminate below the other neurons of the sensillum and form extensive lamellation (*Steinbrecht, 1984*). $DO_{p3}$, $DO_{p5}$, and $DO_{p6}$ each house two neurons. One neuron ends right below the cuticle surface of the sensillum (*Figure 5A and B*) and the other becomes visible within the inner sensillum lymph cavity and forms a lamellated dendritic bulb (*Figure 5E and F*). This remarkable structure might be similar to the 'dendritic bulb' described by recent studies using fluorescence microscopy of transgenic fly lines (*Klein et al., 2015*; *Ni et al., 2016*; *Hernandez-Nunez et al., 2021*). It was shown that the DO neurons associated with these structures trigger cool avoidance behavior in larvae (*Klein et al., 2015*). Presumably, this response is mediated by Ir21a (*Ni et al., 2016*). In addition, two of these sensilla play a role in warmth sensing via Ir68a. This receptor is expressed in the neurons without a dendritic bulb. Both neurons were shown to act together as synchronous and opponent thermosensors (*Hernandez-Nunez et al., 2021*). Contact chemosensory function was assumed for at least two peripheral DO sensilla based on structural characteristics in *Musca* larvae (*Chu and Axtell, 1971*). Further evidence for gustatory function of the DO came from the report of Gr2a and Gr28a gene expression in DO neurons of *Drosophila* larvae (*Colomb et al., 2007*; *Kwon et al., 2011*). $DO_{p2}$ and $DO_{p4}$ each house two neurons, of which one is likely to be mechanosensory. This function is indicated by the presence of a tubular body located at the tip of the neurons' dendrite (*Figure 4D and F*). However, their sensilla structure differs from the gustatory sensilla found in the TO (*Rist and Thum, 2017*). There, dendrites of one gustatory sensillum are surrounded by a common dendritic sheath, which forms the lymph-filled lumen of the sensillum. $DO_{p2}$ and $DO_{p4}$ are of simpler organization, lacking an obvious sensillum lumen as each dendrite is enclosed by an individual dendritic sheath. A cuticle tube, present in all gustatory TO sensilla, is absent, too. Furthermore, a terminal pore is missing in L1 larvae, which argues against a role in contact chemosensation. In adult *Drosophila*, Gr2a and Gr28a have been reported to be expressed not only in gustatory but also in other sensilla of another, yet unknown function (*Thorne and Amrein, 2008*). In addition, Gr28a was also reported to be expressed in the larval gut (*Park and Kwon, 2011*). Consequently, we are questioning the chemosensory roles of DOp2 and DOp4. The structural arrangement of DOp1 and papilla sensillum p6 suggests mechanosensation, as evidenced by the presence of a single neuron forming a tubular body at its dendritic tip (*Figure 4B*). Thus, our primary inference is that p6, DOp1, DOp2, and DOp4 predominantly serve mechanosensory functions, while DOp2B and DOp4B may potentially be associated with a distinct sensory mechanism.

The DO has been reported to consist of 14 sheath cells, 14 shaft cells, and 13 socket cells (*Grillenzoni et al., 2007*). Therefore, it has been concluded that the DO derives from 14 sensory organ precursor cells (SOP) and consists of 14 sensilla. With our data, we could confirm and even extend these assumptions. We hypothesize that the DO consists of 14 sensilla, too, although the 14th sensilla is not the $P_{do}$ but the papilla sensillum $p_6$. Some of its ACs lie outside the DO, which is possibly the reason for the missing 14th socket cell. $P_{do}$ on the other hand, has its sensory cell bodies in the DOG, but all the ACs lie outside the DO.

Furthermore, we could find a total of 60 cells within the DO and a total of 43 cells in the DOG. In one exceptional case, we observed an eighth olfactory triplet in the DO, leading to an altered cell count of three more sensory cells in the DOG and four more potential ACs inside the DO. Therefore, we hypothesize that the olfactory DO sensilla, if not all (DO sensilla), possess four ACs derived from the SOPs. If the fourth one is a second tormogen cell, like in some adult olfactory sensilla (*Shanbhag et al., 2000*; *Nava Gonzales et al., 2021*), or a non-apoptotic glial cell, can't be finally resolved with our data. In addition to the hypothesized 53 ACs ($4 \times 7$ olfactory ACs, $4 \times 6$ peripheral ACs, $1 \times p6$ glial cell) and 36 sensory cells (21 olfactory, 11 peripheral, 1 $p_6$, and 3 $P_{do}$ neurons), we find seven cells

of unknown origin in the DO and in the DOG, respectively. They either also derive from SOPs or, more likely, are peripheral glial cells generated in the CNS that migrate towards their final destination (*von Hilchen et al., 2008*).

## Ventral organ

We conclude that the VO is mainly a mechanosensory organ and its only gustatory sensillum might be $VO_2$. Contrary, a mainly gustatory function of the VO was assumed previously based on the presence of sensillum-terminal pores and the expression of gustatory receptor (Grs) genes in neurons of the VO (*Chu-Wang and Axtell, 1972b*; *Singh and Singh, 1984*; *Python and Stocker, 2002*; *Colomb et al., 2007*). In fact, the terminal pores of most VO sensilla might be traces of molting, as they were absent in L1 larvae of onion and seed-corn flies (*Honda and Ishikawa, 1987*), which have not undergone molting in this stage. Our results validate these findings, as there were no visible terminal pores (*Figure 8C*). VO sensilla $VO_1$, $VO_3$, and $VO_4$ display structural characteristics typical for mechanosensation (*Keil, 1997*): innervation by a single dendrite terminating with a tubular body (*Figure 8C*). Their different size and microtubule organization likely correlate with the processing of different stimulus intensities applied to the mechanotransduction apparatus (*Bechstedt et al., 2010*).$VO_2$, in contrast, displays a morphology similar to the gustatory pit sensilla $T_3$ and $T_4$ of the TO (*Rist and Thum, 2017*). Therefore, $VO_2$ might be the single sensillum of the VO serving contact chemosensation.

## Labial organ

We conclude that the main function of the LO lies in the detection of mechanical stimuli, too. The two single neurons of the LO sensilla each compose a tubular body at their endings (*Figure 9E and F*), indicating mechanosensory function. On the other hand, we cannot exclude additional sensory functions, especially for the $LO_2$, which shows a noticeable outer appearance. It is possible that the club-shaped structure acts in hygrosensation by exerting pressure on the tubular body depending on humidity levels, acting as a hygromechanical transducer (*Tichy and Kallina, 2010*).

## Papilla sensilla

Papilla sensilla is the most abundant type of sensory structure throughout the larval sensory system (*Figure 1B* and *Figure 10A*). They occur in each segment, including the anal division and the head capsule. We conclude that their main function is the detection of mechanical stimuli, given the fact that they all possess a tubular body at the tip of their dendrites (*Figure 10C–G*). Most of the papilla sensilla probably detect mainly deformation of the cuticle by motion or external pressure applied, although the direction of force detection might differ in some papilla, like the abdominal slit papilla $p_5$. The same is true for the abdominal papilla sensilla $p_6$, but they house two sensory cells, $v'es_{2A}$, one of them lacking a tubular body and, therefore, possibly serving a different sensory function.

## Hair sensilla

Hair sensilla is abundant on the thoracic and abdominal segments. Like papilla sensilla, their sensory neurons all end with a tubular body at the dendrite tip, indicating mechanosensation (*Figure 11C, E, F*). As discussed above, hair sensilla most likely transduces directional force applied to the distal end of the hair. In addition to mechanosensory function, another unknown function can be assumed for the abdominal hair sensilla $h_3$. It houses two sensory cells, $des_{2A}$, one of them with and one without a tubular body. The sensory function of these neurons is difficult to assume only by ultrastructural analysis, as they lack obvious structures like pores (gustatory), tubular bodies (mechanosensory), or dendritic bulbs, or lamellated outer dendritic segments (hygro- and thermosensory).

## Knob sensilla

Hygro- and thermosensory function was assumed for knob sensilla (*Hartenstein, 1988*), based on structural similarities with hygro-and thermosensory insect sensilla (*Altner and Prillinger, 1980*; *Altner et al., 1981*; *Steinbrecht, 1984*). Thereafter, knob-shaped sensilla sunken in a cuticle cavity associated with three dendrites typically hint towards a hygro- and or thermosensory function. One of the three neurons typically associated with hygro- and thermosensory sensilla displays peculiar structural characteristics as it might be branched or highly lamellated in its tip that ends beneath

the knob (*Altner and Prillinger, 1980*; *Altner et al., 1981*; *Steinbrecht, 1984*). Instead of lamellation, we find that one neuron of the thoracic knob sensilla contains a tubular body (*Figure 13D, E*). Knob sensilla organized in sensory cones also display one sensory cell containing a tubular body, irrespective if they are innervated by two or three neurons in total (*Figure 16D, E*). This result indicates a mechanosensory and not a hygro- and thermosensory function for these neurons. In the absence of physiological data, however, the sensory function of the knob sensilla remains unclear. A sensory function of the other two knob sensilla neurons in contact chemosensation is not inconceivable. The knob sensilla of the TO are similar in external morphology and are candidates of contact chemosensation due to the indicated expression of certain Grs in their sensory neurons (*Rist and Thum, 2017*). In sensory neurons of thoracic segments, too, expression of Grs (Gr2a) was assumed based on GAL4 driver line analysis (*Colomb et al., 2007*). To which sensillum type these sensory neurons belong can now be determined based on the presently established morphological classification.

Additionally, knob sensilla of the larval thoracic segments were associated with two atypical soluble guanylyl cyclases (asGC), Gyc-89Da and Gyc-89Db, which were shown to sense oxygen and are important for response to hypoxia and in larval ecdysis (*Vermehren-Schmaedick et al., 2010*; *Morton et al., 2008*). Two neurons were found on each thoracic hemisegment which express both subunits, which fit with the number of thoracic knob sensilla. In addition, each subunit was found to be expressed in two different neurons innervating the TO (per side), which also fits the number of knob sensilla in the TO (*Rist and Thum, 2017*). Furthermore, both subunits were co-expressed in neurons innervating the terminal sensory cones. It was stated that this is true for all cones (*Morton et al., 2008*). However, the assigned position seems to be slightly off. We hypothesize that these asGC were actually found in cones containing a knob sensillum ($t_1$, $t_2$, $t_3$, $t_5$, $t_7$). The sixth one, which also appears to be of a different color in the confocal data, is most likely a tracheal dendrite (td) neuron, which we found to occur in this region (not shown). This interpretation is supported by findings that these asGC subunits were also individually expressed in two td neurons in A1 and A2 (per hemisegment) (*Morton et al., 2008*). Because the knob sensilla of the TO are only innervated by one neuron, which protrudes into the knob structure, we conclude that these neurons, in general (for all knob sensilla), are the ones expressing the asGC. Also, within the knob, the dendrites are swollen and not encased by a dendritic sheath, which makes them the most likely candidates for sensing external cues like oxygen (*Figure 13C* and *Figure 16C, G*). Therefore, we conclude that knob sensilla mainly serve chemosensory function (oxygen perception) and mechanosensory function (the latter is not true for TO-associated knob sensilla). $K_2$ in the TO was also associated with CO2 perception, as its single neuron expresses Gr21a and Gr63a (*Rist and Thum, 2017*), which were shown to mediate CO2 responses (*Kwon et al., 2007*; *Kwon et al., 2011*; *Jones et al., 2007*; *Faucher et al., 2006*). In contrast to other larval knob sensilla, the dendrite of $K_2$ is lamellated at its tip, which could also indicate a hygrosensory mechanism (*Altner and Prillinger, 1980*; *Rist and Thum, 2017*). The function of the third neuron innervating the thoracic and some of the knob sensilla of the anal division is not known and will have to be determined in the future.

## Keilin's organ

The KO of *Drosophila* larvae was assumed to serve hygro- and thermosensory functions (*Hafez, 1950*; *Benz, 1956*). However, electrophysiological recordings of the KO of Phormia larvae indicate mechanosensory function, probably associated with the crawling behavior of the larvae (*Lakes-Harlan et al., 1991*). This finding is in line with the anatomical characteristics of the KO. The organ consists of three hair-like and one respectively, two papilla-like sensilla. Each sensory dendrite terminates with a tubular body, which strongly indicates mechanosensory function. The arrangement of the sensory hair sensilla indicates that they are deflected by forces from different directions and, therefore, could be important for orientation. The exploratory behavior of the larva includes straight crawls interspersed with head casts and turns, that the larva executes to redirect the crawling directory. Turns are carried out by backward contractions of the most anterior segments as far as segment four (*Berni et al., 2012*). Therefore, the strongest directional change appears in the thoracic segments, where the KO could sense these changes in direction. In addition, it would also be imaginable that they are involved in the perception of upward and downward movements during orientation behaviors like rearing (*Green et al., 1983*).

## Terminal sensory cones

The terminal sensory cones of the last abdominal segments are specialized sense organs, which house mainly hair and knob sensilla. Their prominent cone-like structures stick out of the food when the larvae are feeding, putting them in an ideal location to sense changes in oxygen concentrations in their surrounding environment (*Vermehren-Schmaedick et al., 2010*). Dwelling into the media enables the larvae to hide from predators and strong daylight to prevent desiccation (*Kim et al., 2017*). But completely submerging into the medium is dangerous as well, as it leads to hypoxia and death, eventually. Therefore, it is essential to dig exactly deep enough to be sheltered but not completely covered. The cone-associated mechanosensory hair sensilla might be activated when the cones are immersed into the medium, and therefore, act as a warning signal for the larva.

## Spiracle sense organ

The sense organ of the posterior spiracle consists of four sensilla that sit around the spiracular opening (*Figure 17A*). They contain one sensory neuron each, with a tubular body at their dendrites' tip (*Figure 17D*), which indicates mechanosensory function. The tubular body is surrounded by a dendritic sheath that is connected to electron-dense cuticular material and the spiracular hairs, which sit on top of the spiracle. The hairs protrude at a 90 degree angle from the spiracle (*Figure 17C, D*) and are probably deflected when the larva submerges into the substrate. Therefore, the spiracle sense organ could be important for the positioning of the spiracle in the right oxygen-rich environment.

## Outlook

In this work, we further extend the understanding of the larval sensory system. We complete the spatial map of all external sensory structures and internal chordotonal organs. We used ultrastructural details to make well-grounded predictions about the putative sensory functions of these sensory organs. These results will serve as a basis for further molecular and functional studies. To complete the understanding of the different modalities of larval sensation, it will be necessary in the future to also address the somatosensory, pharyngeal, and enteric sensory systems with all their sensilla, cells, and ultrastructure. This can now be done in subsequent studies of the EM volume of the entire body of the larva. A complete description of all external and internal sensory inputs of the *Drosophila* larva is now within reach.

# Materials and methods

## Fly husbandry

*Drosophila* wild-type Canton S flies were kept on standard corn meal medium at room temperature, as previously described (*Selcho et al., 2009*; *Rohwedder et al., 2012*; *Apostolopoulou et al., 2014*). For all experiments, third-instar larvae prior to the wandering stage were used.

## Scanning electron microscopy (SEM)

L3 larvae were rinsed with tap water to remove food residuals. Then, specimens were incubated in tap water at 90 °C for 2 min. Subsequently, larvae were fixed in 2.5% glutaraldehyde in 0.1 M HEPES buffer for 3 hr. The fixative was exchanged each hour. Primary fixation was followed by three washing steps in 0.1 M HEPES buffer, each for 10 min. Then, samples were incubated in 1% osmium tetroxide buffered in 0.1 M HEPES for 2 hr, followed by three washing steps in 0.1 M HEPES for 10 min each. All steps were performed at pH 7 and 4°C. Next, samples were gradually dehydrated in an increasing ethanol series: 30%, 50%, 70% two times for 10 min, and then 70% overnight (all at 4 °C). The next day, dehydration was continued with 80%, 90%, three times 96% for 10 min, and 100% four times for 10 min (at room temperature). Dehydrated samples were transferred to a critical point drier (Bal-Tec CPD 030, Liechtenstein) and critical point dried over liquid $CO_2$. Afterward, samples were glued to aluminum SEM-stubs using adhesive conductive carbon pads (Plano GmbH, Germany) or Conduct-C (Conductive Carbon Cement, Plano GmbH) and subsequently coated with ca. 12 nm platinum in a sputter coater (Quorum Q 150 R ES Sputter-Coater, UK). SEM micrographs were acquired on a FESEM Auriga Crossbeam workstation (Zeiss, Germany) using the SmartSEM software package (Zeiss, Germany).

## Focused ion beam-sem (FIB-SEM)

Larvae were rinsed in phosphate-buffered saline (PBS) to remove food residuals. For the study of head organs, the anterior half of the larva was incubated in 2% formaldehyde fixative (Sigma-Aldrich, Germany) with 2.5% glutardialdehyde in 0.1 M Na-cacodylate buffer, pH 7.4 for 30–60 min. Then, the head region was cut off and incubated in fresh fixative for another 90 min. For the investigation of thoracic and abdominal segments, the larva was cut into pieces not larger than four segments. Fixation was carried out similarly to larval heads. Wash- and post-fixation steps were similar to SEM preparation. En-bloc staining was carried out with 1% uranyl acetate and 1% phosphotungstic acid in 70% EtOH in the dark overnight before continuing the alcohol dehydration the next day. Samples were transferred to propylene oxide before embedding in Spurr (Plano, GmbH, Germany) using ascending Spurr concentrations diluted in propylene oxide for optimal tissue infiltration (*Rist and Thum, 2017*). The resin was drained of the samples by gravity during polymerization. Polymerization was carried out at 65 °C for 72 hr. Samples were mounted on aluminum SEM stubs using adhesive conductive carbon pads (Plano GmbH, Germany). Samples were sputter-coated with ~100 nm platinum.

FIB-SEM serial sectioning was carried out using a FESEM Auriga Cross-beam workstation (Zeiss, Germany). Images were acquired using a secondary electron detector or a backscattered electron detector. The three-dimensional FIB-Stack of the DO was acquired using the FIB-SEM operating software package ATLAS 3D Nanotomography (Fibics Inc, Canada).

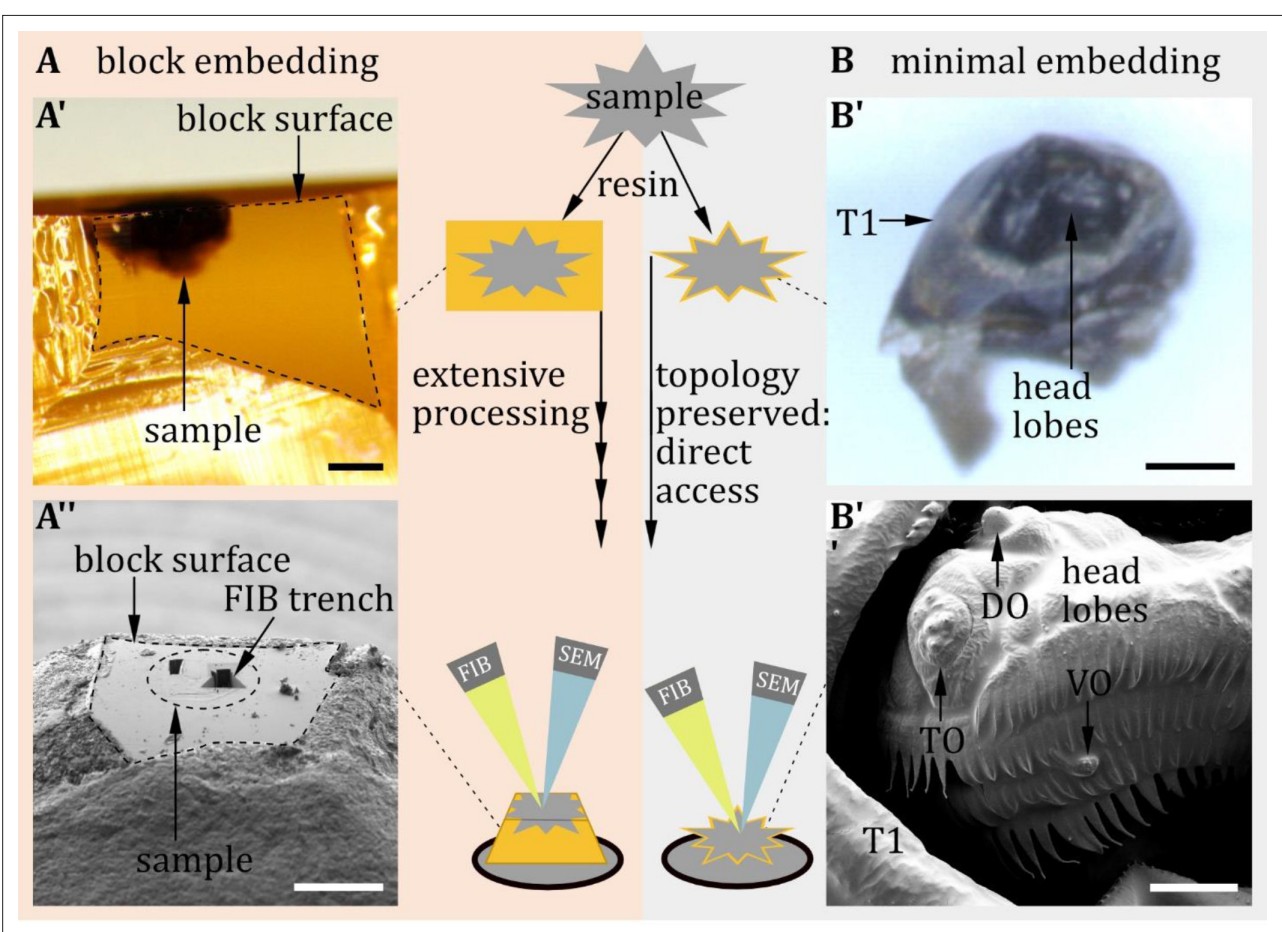

**Figure 18.** Block embedding versus minimal embedding technique. (**A**) In conventional block embedding, the sample is embedded in a bulk of resin. (**A′**) Block-embedded larval head: Frontal view of the block surface imaged with a stereomicroscope. The contours of the cuticle are complex to identify. Therefore, the block leads to extensive processing prior to Focused Ion Beam-SEM (FIB-SEM): excessive resin must be cut away to expose the region of interest to the ion and the electron beam. (**A″**) Block surface seen through the scanning electron microscopy (SEM). Visible are the trenches left by focused ion beam (FIB) slicing of the regions of interest. (**B**) In contrast, minimal embedding preserves the topology of the sample because only a thin layer of resin covers the sample (**B′**). Samples can be glued directly to a SEM stub without prior cutting. Body parts and sensilla are visible by SEM (**B″**) and can be directly targeted by the FIB. Scale bars: (**A′**), (**A″**): 200 μm; (**B′**): 100 μm (**B″**): 20 μm.

In FIB-SEM, layers of the sample are mechanically removed by the ion beam of the FIB. This procedure works only for relatively small volumes. Therefore, samples that are embedded conventionally in a block of resin require cutting off excess resin to expose the tissue of interest to FIB-SEM slicing and imaging. Often, this is very difficult because structures like insect sensilla and sense organs might be too small to be visible through the block surface. Also, the sample lacks contrast as it is dark from the en-bloc-staining with osmium. In this work, we let the samples harden in only minimal amounts of resin instead of embedding the samples in a block of resin (*Schieber et al., 2017*; *Figure 18*). The results of this approach were 'sculptures' of the samples covered only with a very thin layer of resin that left the surface topology of the larva intact. Immediately, these sculptures could be mounted on EM stubs, avoiding further, time-consuming preparation steps. On the minimal embedded sculptures, larger sense organs but also the smallest body wall sensilla could be clearly identified and, therefore, be precisely targeted by FIB-SEM for serial slicing and imaging.

### Whole larval volume

Sensilla and neuron reconstruction was done on a STEM (scanning transmission electron microscopy) volume of a whole first instar larva; information on the technical details of its generation is mentioned in *Peale et al., 2024*.

### IMAGE processing

Out of the whole larval volume, we extracted smaller volumes of the sensory organs. The obtained image stacks were imported to Amira (Thermo Fischer Scientific, v2019). Because the whole larval volume was already aligned, the stacks were only slightly realigned by manual correction. FIB-SEM stacks were aligned in FIJI (*Schindelin et al., 2012*) using the TrakEM2 plugin (*Cardona et al., 2012*). In Amira, structures of interest were segmented and transformed into 3D objects. Next, the segmentations were imported to Blender (Blender Institute, Amsterdam), where the 3D reconstructions were manually finished using the preliminary segmentation as a template.

### FIB-SEM/SEM image processing

Micrographs were processed in FIJI/ImageJ (NIH, USA) for slight contrast and brightness adjustments.

## Acknowledgements

This work was supported by the Deutsche Forschungsgemeinschaft (Grant No. 441181781, 426722269, 432195391), by EU funds from the ESF Plus Program (Grant No. 100649752), and by the Open Access Publishing Fund of Leipzig University supported by the German Research Foundation within the program Open Access Publication Funding. We thank Dennis Pauls, Mareike Selcho, Tilman Triphan, Wolf Hütteroth, Denise Weber, Bert Klagges, and Reini Stocker for their discussions and comments.

## Additional information

### Competing interests

Albert Cardona: Reviewing editor, *eLife*. The other authors declare that no competing interests exist.

### Funding

| Funder | Grant reference number | Author |
| --- | --- | --- |
| Deutsche Forschungsgemeinschaft | 441181781 | Andreas S Thum |
| Deutsche Forschungsgemeinschaft | 426722269 | Andreas S Thum |
| Deutsche Forschungsgemeinschaft | 432195391 | Andreas S Thum |
| European Social Fund Plus | 100649752 | Vincent Richter<br>Andreas S Thum |

| Funder | Grant reference number | Author |
|--------|------------------------|--------|

The funders had no role in study design, data collection and interpretation, or the decision to submit the work for publication.

## Author contributions

Vincent Richter, Conceptualization, Formal analysis, Validation, Investigation, Visualization, Writing – original draft, Project administration, Writing – review and editing; Anna Rist, Conceptualization, Resources, Formal analysis, Investigation, Visualization, Methodology, Writing – original draft; Georg Kislinger, Resources, Investigation, Visualization, Methodology; Michael Laumann, Resources, Methodology; Andreas Schoofs, Resources, Data curation, Supervision, Writing – review and editing; Anton Miroschnikow, Albert Cardona, Resources, Data curation, Supervision, Methodology, Writing – review and editing; Michael J Pankratz, Resources, Supervision, Writing – review and editing; Andreas S Thum, Conceptualization, Supervision, Funding acquisition, Validation, Methodology, Writing – original draft, Project administration, Writing – review and editing

## Author ORCIDs

Vincent Richter ⓘ https://orcid.org/0000-0001-8009-0357
Andreas Schoofs ⓘ https://orcid.org/0000-0001-7002-9181
Anton Miroschnikow ⓘ https://orcid.org/0000-0002-2276-3434
Michael J Pankratz ⓘ https://orcid.org/0000-0001-5458-6471
Andreas S Thum ⓘ https://orcid.org/0000-0002-3830-6596

Reviewer #1 (Public review): https://doi.org/10.7554/eLife.91155.3.sa1
Reviewer #2 (Public review): https://doi.org/10.7554/eLife.91155.3.sa2
Reviewer #3 (Public review): https://doi.org/10.7554/eLife.91155.3.sa3
Author response https://doi.org/10.7554/eLife.91155.3.sa4

# Additional files

## Supplementary files

MDAR checklist

## Data availability

The data supporting this study's findings are available on Dryad at https://doi.org/10.5061/dryad.1vhhmgr1z.

The following dataset was generated:

| Author(s) | Year | Dataset title | Dataset URL | Database and Identifier |
|-----------|------|---------------|-------------|-------------------------|
| Richter V | 2024 | Morphology and ultrastructure of external sense organs of *Drosophila* larvae | https://doi.org/10.5061/dryad.1vhhmgr1z | Dryad Digital Repository, 10.5061/dryad.1vhhmgr1z |

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
