## [Editor Report · eLife assessment]

The manuscript from Richter et al. is a very thorough anatomical description of the external sensory organs in *Drosophila* larvae. It represents a **fundamental** step forward for sensory physiology, and provides a tool for investigating the relationship between the structure and function of sensory organs. Using improved electron microscopy analysis and digital modelling, the authors provide **compelling** evidence that form the basis for further molecular and functional studies to decipher the sensory strategies used by larvae to navigate through their environment.

---

## [Referee Report · Reviewer #1 (Public review)]

Summary: This is a very meticulous and precise anatomical description of the external sensory organs in *Drosophila* larvae. It generates an integral and accurate map. The authors revise all the data for the abdominal and thoracic segments and describe in detail, for the first time, the head and tail segments.

Strengths: It is a very thorough anatomical description of the external sensory organs of the genetically amenable fruitfly. This study represents a very useful tool for the research community that will definitely be used it as a reference paper. It will allow us to investigate sensory processing in depth. The discussion places the anatomical data into a functional and developmental frame.

---

## [Referee Report · Reviewer #2 (Public review)]

Summary: This study is a superbly written and illustrated documentation of the external sensilla of the *Drosophila* larva. Serial electron microscopy and digital modeling is used to the fullest to provide a definitive and clear picture of the sensory organs, which is dearly needed in the field.

Strengths: Serial electron microscopy and digital modeling is used to the fullest to provide a comprehensive, definitive and clear picture of the sensory organs, which is dearly needed in the field.

Weaknesses: none detected.

---

## [Referee Report · Reviewer #3 (Public review)]

Summary: Richter et al. present a comprehensive anatomical analysis of the external sensory organs of the *D. melanogaster* larva. Extending on their previous study (Rist and Thum 2017) that analyzed the anatomy of the terminal organ, a major external taste organ of fruit fly larva, the authors examined the anatomy of the remaining head sensory organs - the dorsal organ, the ventral organ, and the labial organ-also described the sensory organs of the thoracic and abdominal segments. Using improved electron microscopy, the authors performed a three-dimensional anatomical analysis of the sensilla and adjacent ganglia to construct a complete structural and neuronal map of the external larval sensilla.

Strengths: Though the manuscript is lengthy, it is written clearly, and the presented data supports the conclusion. In addition to the classification and nomenclature of the different types of sensilla throughout the larval body, the wealth of data presented here will be valuable to the scientific community. The study offers fundamental anatomical insights, which will be helpful for future functional studies and to understand the sensory strategies of *Drosophila* larvae in response to the external environment. By analyzing different larval stages (L1 and L3), this work offers some insights into the developmental aspects of the larval sense organs and their corresponding sensory cells.

Weaknesses: There are no apparent weaknesses. The repetitiveness of some data and prior studies may be avoided for easy readability.

---

## [Author Response]

The following is the authors’ response to the original reviews.

**eLife assessment**

The manuscript from Richter et al. is a very thorough anatomical description of the external sensory organs in *Drosophila* larvae. It represents an important tool for investigating the relationship between the structure and function of sensory organs. Using improved electron microscopy analysis and digital modeling, the authors provide compelling evidence offering the basis for molecular and functional studies to decipher the sensory strategies of larvae to navigate through their environment.

**Public Reviews:**
SummaryThis is a very meticulous and precise anatomical description of the external sensory organs (sensillia) in *Drosophila* larvae. Extending on their previous study (Rist and Thum 2017) that analyzed the anatomy of the terminal organ, a major external taste organ of fruit fly larva, the authors examined the anatomy of the remaining head sensory organs - the dorsal organ, the ventral organ, and the labial organ-also described the sensory organs of the thoracic and abdominal segments. Improved serial electron microscopy and digital modeling are used to the fullest to provide a definitive and clear picture of the sensory organs, the sensillia, and adjacent ganglia, providing an integral and accurate map, which is dearly needed in the field. The authors revise all the data for the abdominal and thoracic segments and describe in detail, for the first time, the head and tail segments and construct a complete structural and neuronal map of the external larval sensilla.StrengthsIt is a very thorough anatomical description of the external sensory organs of the genetically amenable fruitfly. This study represents a very useful tool for the research community that will definitely use it as a reference paper. In addition to the classification and nomenclature of the different types of sensilla throughout the larval body, the wealth of data presented here will be valuable to the scientific community. It will allow for investigating sensory processing in depth. Serial electron microscopy and digital modeling are used to the fullest to provide a comprehensive, definitive, and clear picture of the sensory organs. The discussion places the anatomical data into a functional and developmental frame. The study offers fundamental anatomical insights, which will be helpful for future functional studies and to understand the sensory strategies of *Drosophila* larvae in response to the external environment. By analyzing different larval stages (L1 and L3), this work offers some insights into the developmental aspects of the larval sense organs and their corresponding sensory cells.WeaknessesThere are no apparent weaknesses, although it is not a complete novel anatomical study. It revisits many data that already existed, adding new information. However, the repetitiveness of some data and prior studies may be avoided for easy readability.

We would like to thank the reviewers for their respective reviews. The detailed comments and efforts have helped us to improve our manuscript. In the following, we have listed the comments one by one and provide the respective information on how we addressed the concerns.

**Recommendations for the authors:**

We have tried to address every single comment as far as possible. In order to structure our response a little better, we have listed the relevant page number and the original comments once again. Directly following this you will find our response and a description of what we have changed in the manuscript.

REVIEWER #1 (Recommendations For The Authors):I have a few comments that will help the reader navigate this long and detailed paper.REVIEWER 1.1. page 4The final section of "the Structural organization of *Drosophila* larvae" needs some reorganization.Specifically:"The DO and the TO are prominently located on the tip of the head lobes" Can the authors rewrite the sentence in a way that it is clear that there is one DO and one VO on each side of the head? Check at the beginning of each section, please. There is a mention about hemi-segments but it is still confusing.

Done – replaced with “The largest sense organs of *Drosophila* larvae are arranged in pairs on the right and left side of the head.”

REVIEWER 1.2. page 5"The sequence of sensilla is always similar for and different between T1, T2-T3, and A1-A7" This sentence is not clear, please break it into two sentences.

Done – replaced with: “We noticed varying arrangements for T1, T2-T3, and A1-A7, with a consistent sequence of sensilla in each configuration.”

REVIEWER 1.3. figures page 4Double hair can't be found in Figure 1B or C (is it h3, h4?) - please clarify.

Done - changed to double hair organ in page 11, included double hair sketch in legend in figure 1B. We changed the name of the structure to double hair organ, to clarify that this is a compound sensillum consisting of two individual sensilla.

REVIEWER 1.4. page 5The authors go back and forth in their descriptions of the different sensory organs. Knob sensilla and then papilla sensilla are discussed and then a few lines later a further description is done. Please unify the description of each separately.

Done – we restructured the whole section.

REVIEWER 1.5. figures page 6"We found three hair sensilla on T1-T3, and "two" on A1-A7" - in the figure there seem to be "four" on A1-A7.

Done – we included the two hair sensilla of the double hair organ

REVIEWER 1.6. figures page 6DORSAL ORGAN:Can the authors explain the colour map meaning in Figure 2A? It is explained in 2C but the image already has colours. Add your sentence "Color code in A applies to all micrographs in this Figure".

Done – we added a sentence to explain that the color code in A applies to the whole figure.

REVIEWER 1.7. page 6Page 10: which comprises seven olfactory sensilla "composing" three dendrites each: replace this with"with". At the end, we want to think 7 X 3 = 21 ORNs.

Done – replaced.

REVIEWER 1.8. page 9CHORDOTONAL ORGANS:"We find these these DO associated ChO (doChO).. .". Please remove one "these"

Done – removed.

REVIEWER 1.9. page 8Is the DO associated ChO part of the dorsal ganglion???? It does not look like it. Could you clarify?

Done – we added a sentence that clarifies that the ChO neuron is not iside the DOG.

REVIEWER 1.10. page 9 VENTRAL ORGAN: A figures page 12Please add to the Figure 8 legend the description of 8c' and 8c'?

Done – added description in figure legend.

B page 98H, what are the *, arrows? Please clarify - it is hard to interpret the figure.

Done – we added parentheses in the figure legend that state which structures the asterisks and arrows indicate.

C page 9"Three of them are innervated by a single neuron () and one by two neurons () (Figure 8F-I). Please add which are innervated by 1 (VO1, VO2-VO4) and which by 2 (VO3).

Done – we added parentheses that clarify which sensilla are innervated by 1 or 2 neurons.

REVIEWER 1.11. page 9Can you add something (or speculate) about the difference in sensory processing of the different types of sensilla?

Done – new sentence in discussion:

‘Their different size and microtubule organization likely correlate with processing of different stimulus intesities applied to the mechanotransduction apparatus (Bechstedt et al. 2010).’

REVIEWER 1.12. figures page 16PAPILLA AND HAIR SENSILLA:FIGURE 10a, please add the name of each sensillum from p1, p2, px py, etc... (if not we have to go back to figure 1 when you describe specific ps.)

Thanks for the comment, it really makes it a lot easier for the reader.

REVIEWER 1.13. figures page 18Figure 11, can you add the name of each hair, please?

Done – updated figure.

REVIEWER 1.14. figures pages 16, 18, 20In Figures 10, 11, and 12 you clearly draw an area on the internal side that I assume is what you call the "electron-dense sheath". It is wider in papilla sensilla than in hair sensilla, most likely due to thedifference in stimuli sensed that you explain in detail in the discussion. Can you say in the figure what this"internal" thing is? Can you add this difference to your list "Apart from the difference in outer appearance and structure of the tubular body"?

This is the basal septum, but it is not certain that it is wider in the papillae sensillae, at least we could not observe this in our data sets. The impression could have been created by different scales in the 3D reconstructions and a perspective view. Therefore, we do not want to list this as a difference here, as we are not sure.

However, we have now specified the socket septum in the figure legends and in Figures 10A, 11A and 12A.

REVIEWER 1.15. page 11KNOB SENSILLA:Page 25;" Knob sensilla have been described under "vaious" names such as": add various.

Done

REVIEWER 1.16. page 12"reveals that the three hair and the two papilla sensilla are associated with a single dendrite." Can you write that "reveals THAT EACH OF the three hair and the two papilla sensilla" if not it seems that there is only one dendrite.

Done

REVIEWER 1.17. figures page 25TERMINAL SENSORY CONES:Please name the t1-t7 cones in Figure 15A.

Done – we updated the figure.

REVIEWER 1.18. page 13The spiracle sense organ deserves a new paragraph. As does the papilla sensillum of the anal plate.

Done – we added subtitles before the prargraphs.

Discussion:REVIEWER 1.19. page 15Page 38: "v'entral" correct typoPAGE 15

Done – we have updated the nomenclature ventral 1 (v), ventral 2 (v’) and ventral 3 (v’’)

**REVIEWER #2 (Recommendations For The Authors):**
I have only a few comments:REVIEWER 2.1. page 5p.5, right column, middle: the use of trichoid, campaniform, and basiconical (sensilla) in previous works were based on even older papers and reviews that attempted to link EM architecture to function (e.g., KEIL, T. A. & STEINBRECHT, R. A. (1986). Mechanosensitive and olfactory sensilla of insects. In Insect Ultrastructure, vol. 2. (ed. R. C. King & H. Akai), pp. 477-516. New York/London: Plenum Press). Trichoid sensilla can be mechano-sensitive, olfactory, or gustatory; trichoid simply refers to the shape (hair). The same applies to basiconical sensilla. The use of "campaniform", which Ghysen et al called "papilla sensilla", was the only really problematic case, because these (*Drosophila* larval) sensilla did not really resemble closely the classical campaniform sensilla (e.g., adult haltere). The only reason we called them campaniform is because they were not more similar to any other type of (previously named) sensillum.

Thank you for the explanation. The nomenclature of structures is generally always a complex topic with often different approaches and principles. We are aware of this and have therefore tried to be as careful as possible. We were not sure from this comment whether you were suggesting to change the text or whether you wanted to explain how these names were assigned to the sensilla in the past. However, we hope that the current version is in line with your understanding, but could of course make changes if necessary (see also comments of reviewer 1).

REVIEWER 2.2. page 9p.21, Labial Organ: the ventral lip is the labium; the dorsal one is the labrum.

Done – replaced labrum with labium.

REVIEWER 2.3. page 9p.20/21, Ventral organ and labial organ: here, the projection of the axons could be mentioned as an ordering principle. In the previous literature, for larva and embryo, a labial organ (lbo) was described that most likely corresponds to the labial organ presented here. This (previously mentioned) lbo characteristically projects along the labial nerve to the labial segment (hence the name). It fasciculates with axons of another sensory complex, also generated by the labial segment, namely the ventral pharyngeal sensory organ (VPS). Does the labial organ described here share this axonal path?

Yes, it has the same axonal pathway and is the same organ as the lbo. We have tried to standardise the nomenclature for all important external head organs (DO, TO, VO, LO) and have therefore used abbreviations with two letters. However, to avoid confusion, we have now added that the LO was also called lbo in the past.

For the ventral organ, the segmental origin (to my knowledge) was never clarified. The axons of the ventral organ project along the maxillary nerve (which carries axons of the terminal=maxillary organ). This nerve, closely before entering the VNC, splits into a main branch to the maxillary segment (TO axons) and a thinner branch that appears to target the mandibular segment. This branch could contain the axons of the ventral organ (as described previously and in this paper). Could the authors confirm this axonal projection of the VO?

In this work, we did not focus on the axonal projections into the SEZ. This is also not a simple and fast process, as in the entire larval dataset, the large head nerves unfortunately exhibit a highly variable quality of representation. Therefore, the reconstruction of nerves and individual neurons within it is often challenging and very time-consuming. The research question is, of course, very intriguing, and one could also attempt to match each sensory neuron of the periphery with the existing map of the brain connectome. However, this is a project in itself, exceeding the scope of this work, and is therefore more feasible as a subsequent project.

**REVIEWER #3 (Recommendations For The Authors):**
Minor suggestions that the authors might consider:REVIEWER 3.1. figures allRecheck the scale bar in figures and figure legends. Missing in a few places.

Done – we replaced or added some (missing) scale bars in figures and figure legends (see annotated figure document).

REVIEWER 3.2. figures page 4The color schematic in Figure 1 can be improved for readability.

Done – we changed the color schematic, especially for the head region to improve readability.